# Male-biased aganglionic megacolon in the *TashT* mouse model of Hirschsprung disease involves upregulation of p53 protein activity and *Ddx3y* gene expression

Tatiana Cardinal[1,2], Karl-Frédérik Bergeron[2,3], Rodolphe Soret[1,2], Ouliana Souchkova[1,2], Christophe Faure[2,4,5], Amélina Guillon[1], Nicolas Pilon[1,2,4]*

1 Molecular Genetics of Development Laboratory, Département des Sciences Biologiques, Université du Québec à Montréal (UQAM), Montréal, Québec, Canada, 2 Centre d'excellence en recherche sur les maladies orphelines–Fondation Courtois (CERMO-FC), Université du Québec à Montréal, Montréal, Québec, Canada, 3 Lipid Metabolism Laboratory, Département des Sciences Biologiques, Université du Québec à Montréal (UQAM), Montréal, Québec, Canada, 4 Département de pédiatrie, Université de Montréal, Montréal, Québec, Canada, 5 Division de gastroentérologie, hépatologie et nutrition pédiatrique, Centre hospitalier universitaire Sainte-Justine, Montréal, Québec, Canada

* pilon.nicolas@uqam.ca

**Data Availability Statement:** All relevant data are within the manuscript and its Supporting Information files.

## Abstract

Hirschsprung disease (HSCR) is a complex genetic disorder of neural crest development resulting in incomplete formation of the enteric nervous system (ENS). This life-threatening neurocristopathy affects 1/5000 live births, with a currently unexplained male-biased ratio. To address this lack of knowledge, we took advantage of the *TashT* mutant mouse line, which is the only HSCR model to display a robust male bias. Our prior work revealed that the *TashT* insertional mutation perturbs a Chr.10 silencer-enriched non-coding region, leading to transcriptional dysregulation of hundreds of genes in neural crest-derived ENS progenitors of both sexes. Here, through sex-stratified transcriptome analyses and targeted overexpression in ENS progenitors, we show that male-biased ENS malformation in *TashT* embryos is not due to upregulation of *Sry*–the murine ortholog of a candidate gene for the HSCR male bias in humans–but instead involves upregulation of another Y-linked gene, *Ddx3y*. This discovery might be clinically relevant since we further found that the DDX3Y protein is also expressed in the ENS of a subset of male HSCR patients. Mechanistically, other data including chromosome conformation captured-based assays and CRISPR/Cas9-mediated deletions suggest that *Ddx3y* upregulation in male *TashT* ENS progenitors is due to increased transactivation by p53, which appears especially active in these cells yet without triggering apoptosis. Accordingly, *in utero* treatment of *TashT* embryos with the p53 inhibitor pifithrin-α decreased *Ddx3y* expression and abolished the otherwise more severe ENS defect in *TashT* males. Our data thus highlight novel pathogenic roles for p53 and DDX3Y during ENS formation in mice, a finding that might help to explain the intriguing male bias of HSCR in humans.

**Funding:** This work was funded by a grant (#MOP-26037) from the Canadian Institutes of Health Research to NP (http://www.cihr-irsc.gc.ca). TC was supported by studentships from the Fonds de la recherche du Québec – Santé (http://www.frqs.gouv.qc.ca) and from the Fondation du grand défi Pierre Lavoie (https://www.fondationgdpl.com). NP is a Senior research scholar of the Fonds de la recherche du Québec – Santé (http://www.frqs.gouv.qc.ca) and the recipient of the UQAM Research Chair on rare genetic diseases (https://uqam.ca). The funders had no role in study design, data collection and analysis, decision to publish, or preparation of the manuscript.

**Competing interests:** The authors have declared that no competing interests exist.

## Author summary

Hirschsprung disease (HSCR) is a life-threatening birth defect characterized by the lack of enteric neurons in the colon. HSCR is due to many gene defects, which are only partially understood. In particular, the reason why more boys than girls are affected by this condition is currently unknown. Yet, a better understanding of HSCR at the genetic/molecular level would help to develop more efficient therapies compared to current surgical approaches. Here, we identified a potential molecular cascade involving sequential upregulation of p53 and DDX3Y that, when inhibited, can eliminate the male bias in a HSCR mouse model. We found DDX3Y to be also specifically expressed in enteric neurons of a subset of male HSCR patients, suggesting that manipulation of the putative p53-DDX3Y pathway might eventually have therapeutic value in humans as well.

## Introduction

Hirschsprung disease (HSCR), also known as aganglionic megacolon, is a severe male-biased congenital malformation where ganglia of the enteric nervous system (ENS) are missing from the rectum and over a variable length of the distal bowel [1, 2]. As the ENS normally controls patterns of smooth muscle contraction/relaxation along the entire gastrointestinal tract, the lack of ENS ganglia (aganglionosis) in the rectum and distal colon leads to functional obstruction and massive accumulation of fecal material in the proximal colon (megacolon). If left untreated, HSCR is lethal mostly because affected children eventually develop enterocolitis and sepsis [2]. Current treatment of HSCR is surgical removal of the distal aganglionic segment followed by anastomosis between a proximal region with ENS ganglia and the anal verge. Although this approach is usually life saving, there is a great need for alternative treatments since many of these children continue to have severe problems after surgery [3]. This calls for a better understanding of the pathogenic mechanisms underlying HSCR, which are not fully defined. The male bias of HSCR is especially poorly understood.

HSCR occurs because of incomplete colonization of the developing intestines by ENS progenitors–hereinafter named enteric neural crest cells (ENCCs) [4–6]. The vast majority of these progenitors are derived from neural crest cells of vagal origin [7, 8], with minor contingents also provided by neural crest cells of sacral origin [9] and by neural crest-derived Schwann cell precursors from extrinsic nerves [10]. In the mouse, ENCCs of vagal origin initially colonize the foregut around embryonic day (e) 9.5 and then migrate in the anteroposterior direction to reach the anal region by e14.5. From this stage onward, ENCCs of sacral and Schwann cell origin also enter the hindgut, but they cannot compensate for a lack of ENCCs of vagal origin [11]. For this reason, anything that perturbs colonization of the bowel by ENCCs of vagal origin results in aganglionosis over a varying length of the colon. Yet, for most HSCR patients, aganglionosis is restricted to the rectosigmoid colon [2]. This so-called short-segment form is also where the sex bias is more clearly observed, being about four times more common in boys than in girls [1].

The genetics of HSCR is complex, and far from being well understood. Numerous genes have been described to be involved, and most recent studies suggest that different combinations of rare and common variants in these genes are needed for the disease to occur [12–15]. Among this constantly expanding set of genes, those that code for members of GDNF/RET and EDN3/EDNRB signaling pathways have caught attention the most. However, mutations in these genes or in any other previously reported HSCR-associated genes cannot currently explain the strong male sex bias observed in the clinic. Accordingly, this male sex bias has been historically poorly replicated in animal models. A weak male bias has only occasionally been

detected in single and double mutant rodents bearing hypomorphic and/or dominant negative alleles of *Ret* and *Ednrb* [16–20]. In these models, the male bias affects extent of aganglionosis and rarely impacts megacolon expressivity, with the exception of a modest male to female ratio of about 1.5:1 in short-segment aganglionosis [17, 18]. A potential mechanism underlying these observations in rodents could be that the EDNRB pathway is by default slightly less active in the developing male bowel because of reduced expression of both *Edn3* and *Ece1* genes [20].

Other mechanisms that might explain the male bias of HSCR include downregulation of an X-linked gene or upregulation of a Y-linked gene [4]. For instance, a subset of patients suffering from CRASH syndrome-associated loss-of-function mutations of the X-linked gene *L1CAM* are known to also display HSCR [21–23]. However, the estimated incidence of HSCR among those patients is very low (~3%) and, in mouse models, *L1cam* has been shown to act as a modifier gene for aganglionosis-causing mutations of *Sox10* but not for mutations of the major HSCR-associated genes *Ret* and *Gdnf* [24]. More recently, the Y-linked gene *SRY* also emerged as a possible candidate. Although this gene is best known as the key genetic switch for sex determination in the mammalian genital ridge [25], it is also believed to be involved in the etiology of diverse male-biased neurological disorders such as Parkinson's disease and autism [26, 27]. In the context of HSCR, the regulatory sequences of human and pig *SRY* can drive reporter gene expression in the murine neural crest lineage [28, 29], and the SRY protein has been reported to be specifically expressed in the postnatal ENS from the proximal colon of a subset of male HSCR patients [30]. *In vitro* studies further revealed that human SRY can repress the *RET* promoter by preventing the formation of a SOX10-containing transcriptional complex [30]. However, transcriptome analyses of wild-type murine ENCCs show a complete absence of endogenous *Sry* expression [29, 31, 32]. Whether *Sry*/*SRY* expression may be upregulated in the developing ENS under certain circumstances, and whether such upregulation may play a causative pathological role *in vivo* both remain unanswered questions.

The *TashT* insertional mutant mouse line is a reliable model of short-segment HSCR that faithfully replicates key hallmarks of the human disease, such as partial penetrance of the megacolon phenotype [1, 29] and presence of an abnormal ENS (*i.e.*, reduced neuronal density and increased proportion of nitrergic neurons) upstream of the aganglionic segment [33, 34]. Similar ENS defects are also observed in other models of short-segment HSCR like *Ednrb*-null and *Holstein* (a model for Trisomy 21 [Collagen VI]-associated HSCR) mouse lines [32, 35], and all three models show a similar dysbiotic gut microbiome profile mainly characterized by a robust increase of *Proteobacteria* at the expense of *Firmicutes* [36–38]. Yet, the *TashT* line is the only model that displays a robust male bias, not only for extent of colonic aganglionosis but also for megacolon expressivity [29, 34, 37]. Aganglionosis occurs in homozygous *TashT$^{Tg/Tg}$* animals of both sexes due to slower migration of ENCCs [29]. This defect is slightly exaggerated in males, eventually culminating in a sex-associated difference of about 10–15% in length of ENS-covered colon that is clearly visible at birth [29]. In the end, megacolon is frequently observed in *TashT$^{Tg/Tg}$* males and rarely observed in *TashT$^{Tg/Tg}$* females, because more males than females fail to reach a threshold level of minimal intrinsic innervation of the colon, which is estimated to be around 80% of total colon length in the FVB/N genetic background [29, 37].

The *TashT* line has been obtained through an insertional mutagenesis screen for new genes important for neural crest development [39]. The *TashT* transgene insertion site is located in the vicinity of a block of three conserved silencer elements within the *Hace1-Grik2* intergenic region on chromosome 10, resulting in massive dysregulation of the ENCC transcriptome [29]. More specifically, the mutagenic *TashT* insertion was found to disrupt a far-*cis* chromatin interaction normally occurring between this silencer-enriched region and *Fam162b* located at 3.6Mb away, presumably explaining why this poorly characterized gene is upregulated in *TashT$^{Tg/Tg}$* ENCCs [29]. However, neural crest-specific overexpression of *Fam162b* via

transient transgenesis only had a modest negative impact on extent of ENCC colonization in the distal colon [29], suggesting that other gene(s) might also be primary target(s) of the *TashT* insertional mutation, and might thus also contribute to the *TashT*$^{Tg/Tg}$ phenotype.

Here, we further analyzed *TashT*$^{Tg/Tg}$ ENCCs as a function of animal sex, with the main goal of identifying the molecular cause of the *TashT* male bias. Mainly based on unbiased high-throughput approaches and new genetically-modified mouse models, this work led to the identification of the Y-linked gene *Ddx3y* as the likely causal gene of the *TashT* male bias. This work further suggests that sequential upregulation of p53 protein activity and *Ddx3y* gene expression is a reversible pathogenic mechanism in the *TashT* ENS, and correlative expression data show that such mechanism might also potentially be at work in human HSCR.

## Results

### *Ddx3y* is a prime candidate for explaining the male bias of aganglionic megacolon occurrence in *TashT* mice

To identify the molecular cause for the male bias in *TashT*$^{Tg/Tg}$ animals, we first analyzed global gene expression in *TashT*$^{Tg/Tg}$ ENCCs as a function of sex. As per our prior work [29], *TashT*$^{Tg/Tg}$ ENCCs were recovered from e12.5 stomachs and small intestines by FACS (fluorescence-activated cell sorting) owing to the presence of a *SRY* promoter-driven fluorescent reporter (*pSRYp[1.6kb]-YFP*) that specifically labels neural crest derivatives [28]. Prior to cell sorting, embryos were sexed by visual inspection of the dimorphic developing gonads, which are easy to distinguish at e12.5 thanks to the presence of testis-specific features such as the coelomic vessel and testis cords. Pools of sexed ENCC samples were then processed for rRNA-depleted RNA-seq, and sequencing results were analyzed to detect differentially-expressed genes. Very interestingly, comparison of this novel *[Male]TashT*$^{Tg/Tg}$-*vs-[Female]TashT*$^{Tg/Tg}$ dataset (8 genes differently expressed >1.5-fold at *P*≤0.05) with our previously described [29] unsexed *TashT*$^{Tg/Tg}$-*vs*-Control dataset (2075 genes differently expressed >1.5-fold at *P*≤0.05) identified the Y-linked *Ddx3y* as the only gene significantly dysregulated in both analyses (Fig 1A and 1B and S1 and S2 Datasets). This DEAD-box RNA helicase coding gene is upregulated in male *TashT*$^{Tg/Tg}$ ENCCs, and together with three neighbour genes from the short arm of the Y chromosome (*Kdm5d*, *Eif2s3y*, and *Uty*) define the Y-linked gene expression signature of ENCCs [29]. Once again, *Sry* expression was not detected in this transcriptome analysis (S1 Dataset). We also noted apparently robust basal expression of *Ddx3y*, *Eif2s3y*, *Uty* and *Kdm5d* in the female *TashT*$^{Tg/Tg}$ dataset (Fig 1B), most likely due to erroneous mapping of sequencing reads from their highly similar X-linked homologs (*Ddx3x*, *Kdm5c*, *Eif2s3x* and *Utx*). RT-qPCR validation of RNA-seq data in male e12.5 ENCCs recovered by FACS was thus performed using primer pairs specific to Y-linked homologs, which confirmed that *Ddx3y*, but not its neighbours *Uty* and *Eif2s3y*, is specifically upregulated in male *TashT*$^{Tg/Tg}$ ENCCs in comparison to *Gata4p-GFP* control ENCCs [29, 40] of the same sex (Fig 1C and 1D). Although a fully sex-stratified RNA-seq approach (*i.e.*, comparing *TashT*$^{Tg/Tg}$ males and females to control males and females, respectively) might potentially have identified additional genes, all of the above gene expression data strongly suggest that specific upregulation of *Ddx3y* in male *TashT*$^{Tg/Tg}$ ENCCs might contribute to the male bias in extent of aganglionosis, and megacolon expressivity subsequently observed in postnatal animals.

### Overexpression of *Ddx3y*, but not *Sry*, plays a pathogenic role in male *TashT*$^{Tg/Tg}$ ENCCs

To assess a potential pathogenic role for *Ddx3y* overexpression in ENCCs, we generated transgenic mice in which a Myc-tagged version of DDX3Y is expressed as part of a bicistronic

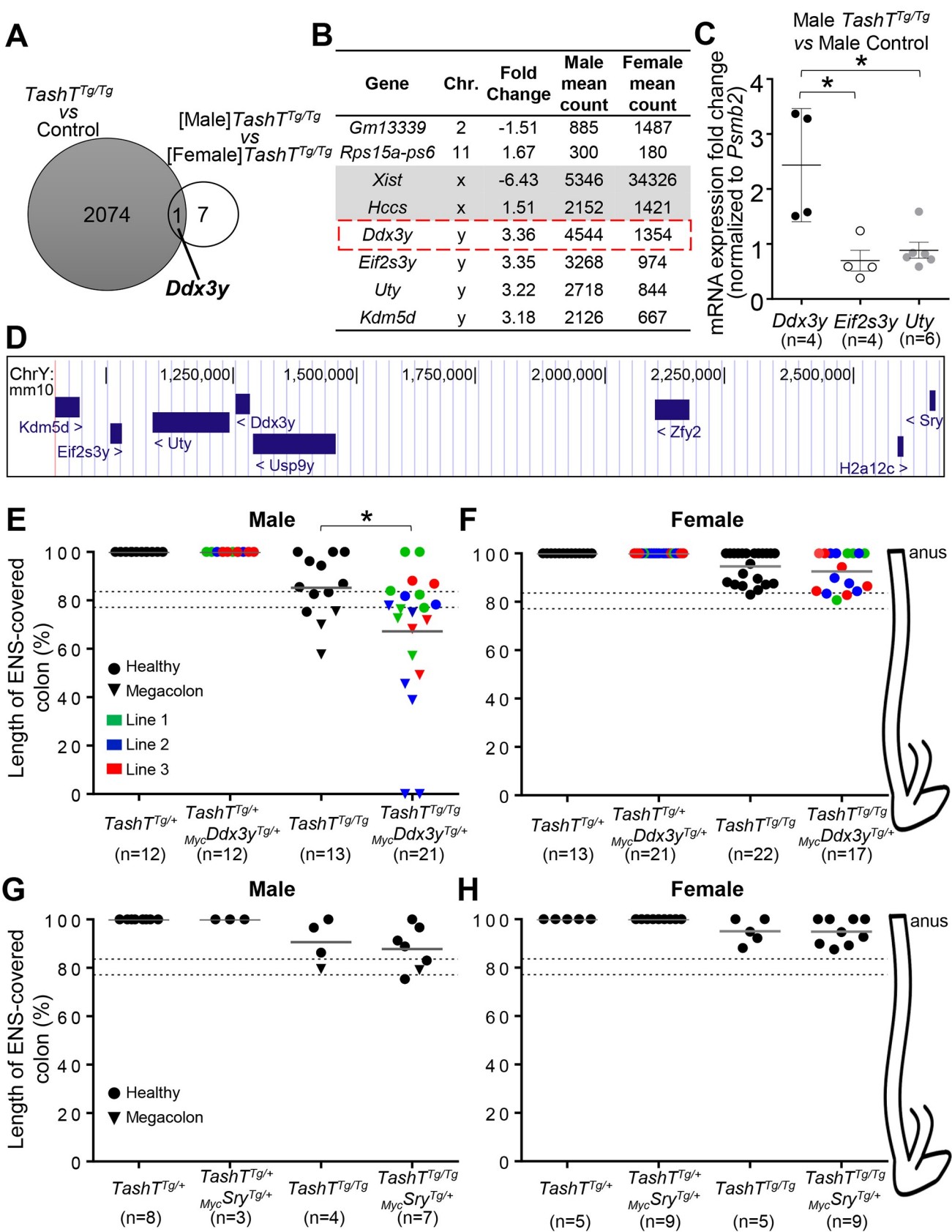

**Fig 1.** *Ddx3y* **upregulation plays a role for the male bias of the** *TashT* **megacolon phenotype.** (A) Venn diagram highlighting that *Ddx3y* is the only gene dysregulated in both unsexed *TashT*$^{Tg/Tg}$-*vs*-Control (S1 Dataset) and *[Male]TashT*$^{Tg/Tg}$-*vs*-*[Female]TashT*$^{Tg/Tg}$ (S2 Dataset) RNA-seq datasets, which were generated from FACS-recovered e12.5 ENCCs (>1.5-fold change; $P \leq 0.05$; minimum of 250 reads in at least one condition). (**B**) List of the 8 dysregulated genes from the *[Male]TashT*$^{Tg/Tg}$-*vs*-*[Female]TashT*$^{Tg/Tg}$ dataset, with *Ddx3y* framed in red. (**C**) *Psmd2*-normalized RT-qPCR expression analysis of *Ddx3y* and its neighbour genes *Eif2s3y* and *Uty* in male e12.5 ENCCs recovered by FACS, showing specific upregulation of *Ddx3y* in male *TashT*$^{Tg/Tg}$ cells compared to male *Gata4p-GFP* control cells. (**D**) Schematic view of the *Ddx3y* locus and its neighbour genes on the short arm of the murine Y chromosome (adapted from the UCSC genome browser; genome.ucsc.edu). (**E-H**) Sex-stratified quantitative analysis of the length of ENS-covered colon (in % of total colon length) in P18-P50 *TashT*$^{Tg/+}$ and *TashT*$^{Tg/Tg}$ mice bearing or not a $_{Myc}$*Ddx3y* transgenic allele (**E-F**) or a $_{Myc}$*Sry* transgenic allele (**G-H**). The dashed lines show the previously described threshold level interval above which megacolon is less likely to occur in FVB/N mice [29]. Results shown in panel E are also independently displayed for each line in S1E Fig. Representative images of colon tissues stained for AChE activity are shown in S4 and S5 Figs. (* $P \leq 0.05$; Student's *t*-test).

construct also allowing expression of eGFP (enhanced green fluorescent protein). To direct expression of this $_{Myc}$*Ddx3y-IRES-eGFP* cassette in ENCCs, we constructed a novel synthetic promoter consisting of fused *Sox10* [41] and *Ednrb* [42] neural crest enhancers upstream of the *Hsp68* minimal promoter [43]. Having previously used a similar *Sox10-U3* enhancer-based system but with mitigated success [29, 31], we reasoned that adding the ENCC-specific *Ednrb* enhancer would increase expression levels and specificity (S1A and S1B Fig). Activity of this novel synthetic promoter and expression of the $_{Myc}$*Ddx3y-IRES-eGFP* cassette were tested in neural crest-derived Neuro2a cells, which confirmed proper cytoplasmic enrichment of $_{Myc}$DDX3Y protein [44] and co-expression with eGFP (S1C Fig). Through standard transgenesis, three founder animals were then obtained and used to generate independent lines. For each line, intestines from e10.5–12.5 transgenic embryos were first screened for eGFP fluorescence, which was undetectable in all cases. Expression of the $_{Myc}$*Ddx3y-IRES-eGFP* transgene in e12.5 ENCCs was nonetheless confirmed for all three lines via RT-qPCR, which further revealed variable levels of expression in each line (S1D Fig). Moreover, we confirmed expression of $_{Myc}$DDX3Y protein via anti-Myc tag immunofluorescence staining of freshly dissociated e12.5 intestines (see results for line #3 in S2 Fig). For each line, both heterozygous ($_{Myc}$*Ddx3y*$^{Tg/+}$) and homozygous ($_{Myc}$*Ddx3y*$^{Tg/Tg}$) transgenic animals were viable and fertile, without any sign of megacolon in rodents (*e.g.*, hunched posture, lethargy, distended abdomen). Accordingly, staining of acetylcholinesterase (AChE) activity in the myenteric plexus revealed an overtly normal ENS structure in each line, regardless of animal sex (S3 Fig). However, in a *TashT*$^{Tg/Tg}$ sensitized background, heterozygous presence of the $_{Myc}$*Ddx3y-IRES-eGFP* transgene from any of the transgenic lines increased megacolon incidence in P18-P50 males (Fig 1E and S1E Fig). The proportion of affected animals reached 37.5% (3/8) for line#1, 75% (6/8) for line #2, and 60% (3/5) for line #3, collectively resulting in a 2.5-fold increase in megacolon incidence from 23% (3/13) of *TashT*$^{Tg/Tg}$ males to 57% (12/21) of all male *TashT*$^{Tg/Tg}$;$_{Myc}$*Ddx3y*$^{Tg/+}$ mice. This collective increase in megacolon incidence was associated with more extensive aganglionosis as assessed via staining of AChE activity in the myenteric plexus (Fig 1E). Each individual line showed an increase in the average extent of aganglionosis, although statistical significance was not reached for lines #1 and #3 most likely because of small sample size (S1E Fig). Nonetheless, extent of aganglionosis was in each case sufficient to markedly increase the proportion of animals not reaching the minimal length of intrinsic innervation necessary to avoid blockage (~80% in the FVB/N genetic background [29]; see dashed lines in Fig 1E and S1E Fig). Intriguingly, neither megacolon incidence nor extent of aganglionosis were affected by the presence of $_{Myc}$DDX3Y in female *TashT*$^{Tg/Tg}$ animals (Fig 1F and S5 Fig).

Based on the results above, we reasoned that this approach would be ideal to also test the candidacy of *Sry* as another Y-linked gene potentially involved in the male bias of HSCR [30]. To address the lack of *in vivo* evidence about a causative pathogenic role for *Sry* in the

developing ENS, we substituted $_{Myc}Ddx3y$ for $_{Myc}Sry$ in the bicistronic construct described above (S6A Fig) and tested the impact of its overexpression in the $TashT^{Tg/Tg}$ sensitized background. As above, the $_{Myc}Sry$-IRES-eGFP construct was first tested in Neuro2a cells, which confirmed proper nuclear localization of $_{Myc}$SRY protein [45] and co-expression with eGFP (S6B Fig). Standard transgenesis then yielded a single founder animal that was used to derive a stable line, in which we confirmed expression of the $_{Myc}Sry$-IRES-eGFP cassette in the developing peripheral nervous system (including the ENS) by virtue of the presence of eGFP fluorescence in e11.5 embryos (S6C Fig). Again, neither $_{Myc}Sry^{Tg/+}$ nor $_{Myc}Sry^{Tg/Tg}$ animals showed a megacolon phenotype. Furthermore, in contrast to our findings with the $_{Myc}Ddx3y$-IRES-eGFP transgene, expression of the $_{Myc}Sry$-IRES-eGFP construct in the $TashT^{Tg/Tg}$ background had no impact on megacolon incidence, extent of aganglionosis or ENS patterning in P18-P50 mice, regardless of animal sex (Fig 1G and 1H and S3–S5 Figs). This work thus suggests that $_{Myc}$SRY is unable to significantly affect murine ENS development *in vivo*, even when expressed in a sensitized genetic background and at higher levels than $_{Myc}$DDX3Y (based on detection of eGFP fluorescence, which further revealed extensive overlap with SOX10$^+$ cells; S7 Fig). Another conclusion of these studies is that only a modest increase of DDX3Y amounts appears sufficient to impair male ENS development in a sensitized genetic background.

## DDX3Y is expressed in human enteric neurons from a subset of male HSCR patients

To verify whether our findings in mice might be clinically relevant, we then evaluated whether the DDX3Y protein is expressed in colon tissues resected from human HSCR patients with short-segment aganglionosis. Although DDX3Y would be expected to have a pathogenic role before birth (*i.e.*, during ENS formation), we reasoned that at least some residual expression might still be detectable in male HSCR tissues a few weeks/months after birth. DDX3Y expression was assessed by immunofluorescence in two distinct cohorts of HSCR patients, including one previously described [32]. A total of 16 tissues were analyzed (14 from boys and 2 from girls), focusing on the ENS-containing region in the intermediate zone just upstream of the aganglionic region, and using the pan-neuronal marker PGP9.5 to identify enteric neurons. As expected, DDX3Y was not detected in enteric neurons from female samples, which were used as negative controls (Fig 2 and Table 1). In marked contrast, variable levels of DDX3Y protein were detected in enteric neurons from half of male patients (Fig 2 and Table 1). Although a definitive answer about the exact proportion of male patients overexpressing DDX3Y would require an analysis of prenatal HSCR tissues (which is impracticable given that HSCR is diagnosed only after birth), these observations in postnatal tissues support the possibility that DDX3Y might play a pathogenic role in the male bias of human HSCR, as it does in the *TashT* mouse line.

## *Ddx3y* expression is not directly regulated by the *Hace1-Grik2* silencer-enriched region in mice

Having discovered that *Ddx3y* upregulation is likely playing a role in the *TashT* megacolon male bias and that the corresponding protein might also potentially contribute to human HSCR, we next sought to determine the cause of this upregulation assuming that the *TashT* insertion site was somehow involved. This transgenic insertion is located within the *Hace1-Grik2* intergenic region on chromosome 10, interfering with three conserved silencer elements [29]. As this silencer-enriched region can normally interact with remote genomic regions of chromosome 10 [29], we first hypothesized that they could also directly interact with the *Ddx3y* locus and thereby actively attenuate its expression during normal ENS formation in

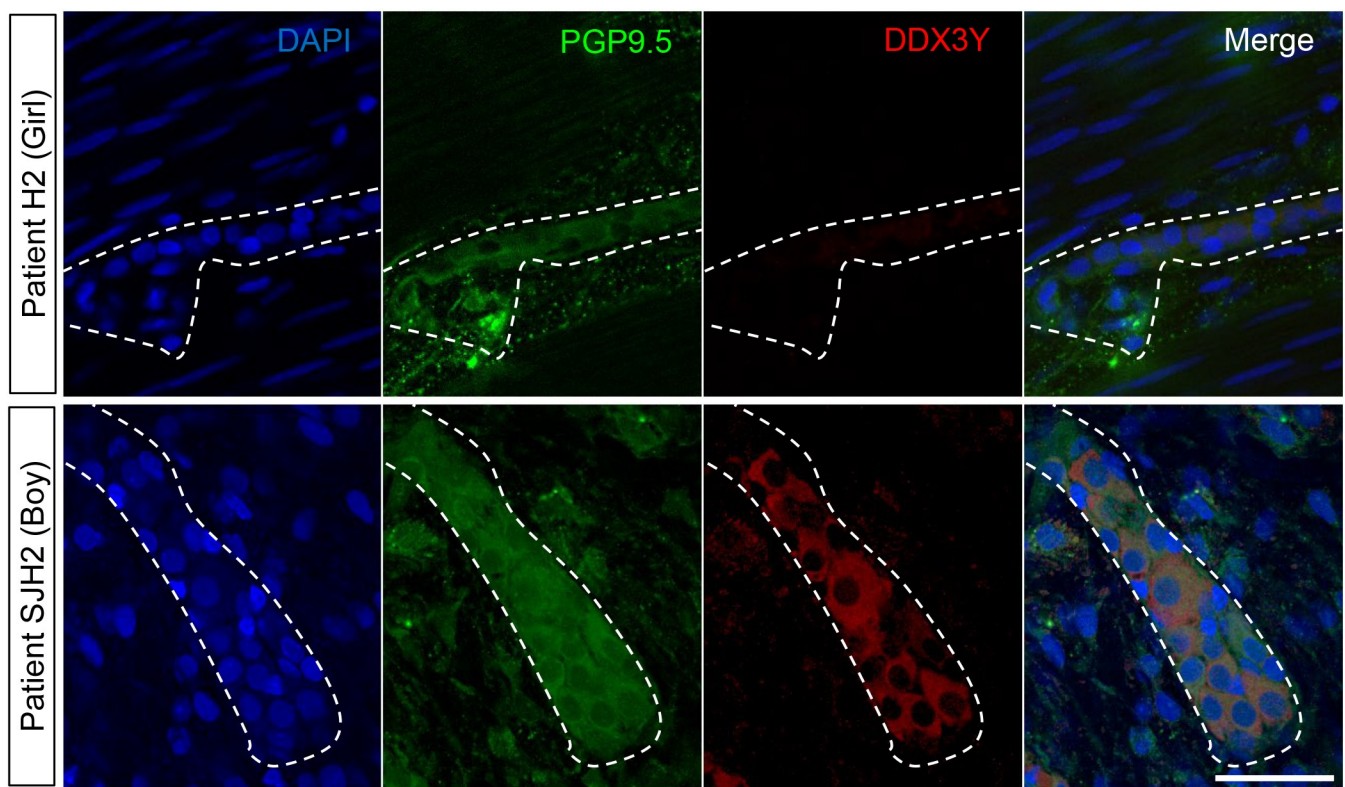

**Fig 2. The DDX3Y protein is present in enteric neurons from the intermediate zone in a subset of HSCR patients.** Representative confocal images of DDX3Y and PGP9.5 double immunofluorescence labeling of enteric neurons in sigmoid colon tissues resected from female (ID #H2) and male (ID #SJH2) HSCR patients. All displayed images represent a single focal plane and dashed outline marks area occupied by a single ganglion. Scale bar, 50μm.

**Table 1. Overview of HSCR colon samples used for assessing DDX3Y expression.**

| Cohort | ID | Age at surgery (days) | Sex | DDX3Y expression |
|---|---|---|---|---|
| Soret et al. [32] | H11 | 35 | F | - |
| | H2 | 75 | F | - |
| | H1 | 17 | M | - |
| | H8 | 20 | M | + |
| | H5 | 40 | M | - |
| | H12 | 42 | M | - |
| | H4 | 51 | M | + |
| | H10 | 63 | M | - |
| | H7 | 96 | M | + |
| | H9 | 222 | M | + |
| | H3 | 321 | M | - |
| | H6 | 454 | M | - |
| Ste-Justine Hospital | STH1 | 22 | M | + |
| | STH2 | 36 | M | + |
| | STH3 | 80 | M | - |
| | STH4 | 349 | M | + |

male embryos. A corollary of this hypothesis would be that the *TashT* transgenic insertion prevents this inter-chromosomic interaction, leading to relief of *Ddx3y* repression in male *TashT*<sup>Tg/Tg</sup> ENCCs. To verify this possibility and potentially identify other relevant interactions with the *Hace1-Grik2* silencer-enriched region, we turned to a circular chromatin conformation capture–high-throughput sequencing (4C-seq) approach [46]. PCR-based pilot assays were first performed in Neuro2a cells in order to test several potential bait sequences located within or around the silencer-enriched region (S8 Fig), from which we chose a 481bp region located between two silencer elements that fulfilled all selection criteria for 4C-seq [46]. To allow comparison with our prior RNA-seq data, the 4C-seq analysis was then performed on FACS-recovered e12.5 ENCCs pooled from either *Gata4p-GFP* control or *TashT*<sup>Tg/Tg</sup> mutant embryos, again as a function of sex. Given that none of the dysregulated genes from *TashT*<sup>Tg/Tg</sup> ENCCs is in the immediate vicinity of the *Hace1-Grik2* intergenic region [29], analysis of captured sequences was focused on far-*cis* (>10kb) and *trans* interactions using a modified version of the W4Cseq pipeline [47]. This analysis did identify chromatin contacts between the *Hace1-Grik2* silencer-enriched region and the *Ddx3y* locus (Fig 3A and S3 Dataset). However, such interactions appeared to be present in both *Gata4p-GFP* control and *TashT*<sup>Tg/Tg</sup> ENCCs, irrespective of sex–most likely because of erroneous mapping of homologous sequences from the X chromosome as observed in sex-stratified RNA-seq data (Fig 1B). This result thus invalidated our primary hypothesis that *Ddx3y* upregulation in male *TashT*<sup>Tg/Tg</sup> ENCCs could be caused by transgene insertion-mediated disruption of a direct functional interaction with the silencer-enriched region. Yet, many gains and losses of chromatin interactions were identified when comparing *Gata4p-GFP* control and *TashT*<sup>Tg/Tg</sup> ENCCs as a function of sex (S4 Dataset), suggesting the possibility of an indirect regulatory mechanism.

## Deletion of the *Hace1-Grik2* silencer-enriched region does not cause colonic aganglionosis in mice

Since our 4C-seq analysis identified many gains and losses of chromatin contacts with the *Hace1-Grik2* silencer-enriched region, we surmised that the *TashT* transgene insertion event causing the megacolon phenotype interferes with normal chromatin interactions and/or triggers new ectopic interactions. To directly test if interaction losses contribute to the megacolon phenotype, we generated mice lacking a 23kb genomic fragment that spans the whole silencer-enriched region from the *Hace1-Grik2* intergenic region (Fig 3B). Through CRISPR/Cas9 targeting, seven founder animals were obtained from which three were used to generate independent lines. In each case, mice homozygous for the deletion (*Hace1-Grik2*<sup>ΔSER/ΔSER</sup>) were viable and fertile. *Hace1-Grik2*<sup>ΔSER/ΔSER</sup> mice also displayed a normally formed ENS in both male and female animals, as assessed in adults via staining of AChE activity in the myenteric plexus (Fig 3C and S9A Fig) and in e12.5 embryos via βIII-Tubulin immunofluorescence staining of ENCCs (Fig 3D and S9B Fig). This negative result thus suggests that neither the *TashT* megacolon phenotype nor its male bias are caused by a transgenic insertion-induced loss of chromatin interaction. This further suggests that the *TashT* male-biased megacolon phenotype is instead due to a transgenic insertion-induced gain of chromatin interaction.

## *Ddx3y* upregulation in male *TashT*<sup>Tg/Tg</sup> ENCCs may be explained by a gain of interaction between the Hace1-Grik2 silencer-enriched region and the p53 phosphatase coding gene *Dusp26*

To explore the contribution of ectopic transgenic insertion-induced chromatin interactions to the male-biased *TashT* megacolon phenotype, we cross-compared our four 4C-seq datasets and focused our attention on 455 genes with which chromatin interaction gains were observed

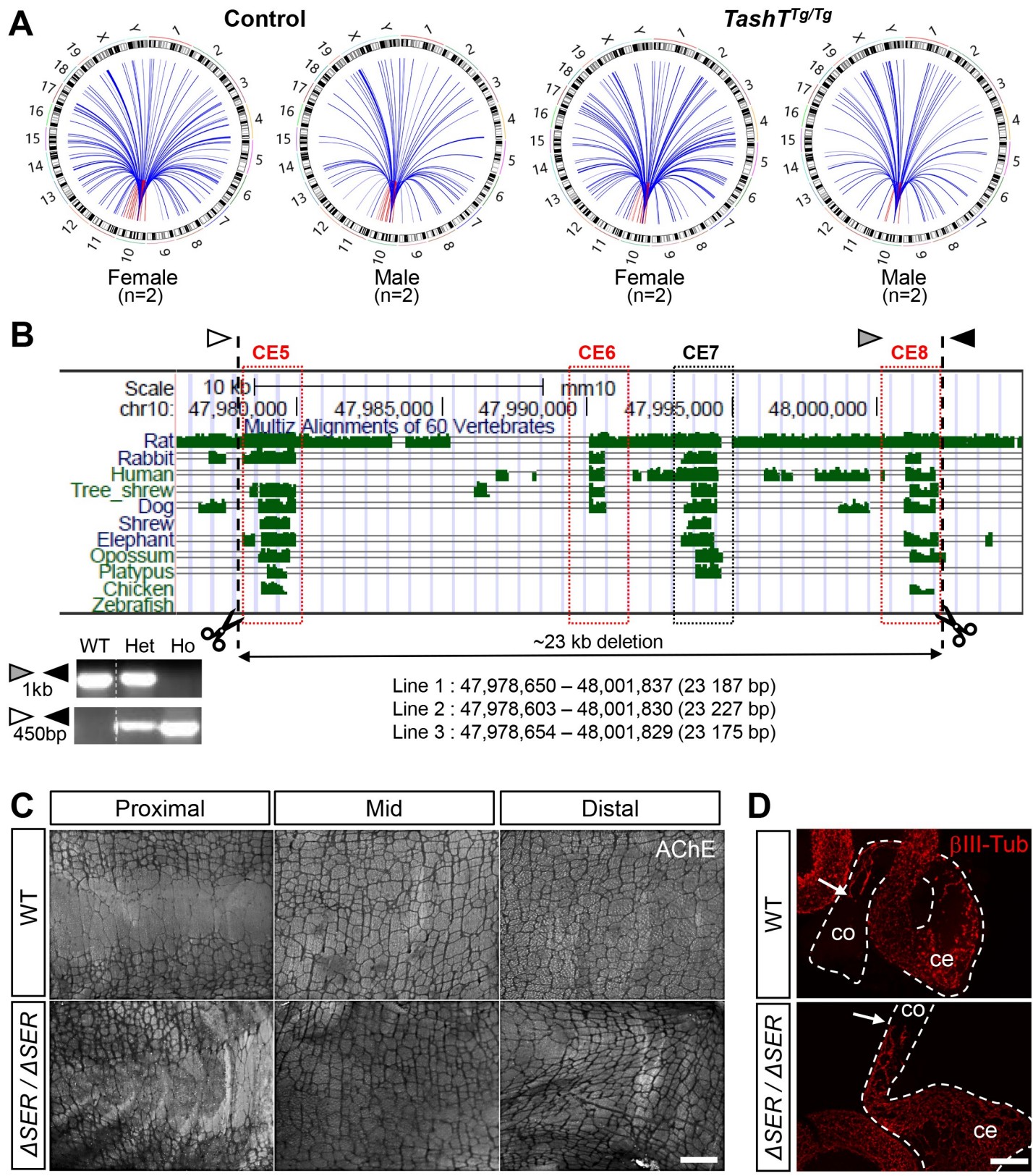

**Fig 3. *Ddx3y* expression is not directly regulated by the *Hace1-Grik2* silencer-enriched region on mouse chromosome 10.** (A) Overview of sex-stratified 4C-seq analysis of far-*cis* (red) and *trans* (blue) interactions with the Chr.10 *Hace1-Grik2* silencer-enriched region in e12.5 ENCCs, which were recovered by FACS from

*Gata4p-GFP* control and *TashT*^Tg/Tg^ embryos. (**B**) Graphical view of part of the *Hace1-Grik2* intergenic region showing the relative position of five evolutionary conserved elements (CE). Those that were previously shown to possess silencer activity [29] are highlighted in red. Genomic positions of the gRNAs used for CRISPR-mediated deletion of a 23kb fragment are indicated by dashed lines. Exact deletion size is indicated for each of the three founder animals that were used to derive independent mouse lines, which were named *Hace1-Grik2*^ΔSER/ΔSER^ (*ΔSER/ΔSER*). Arrowheads (white, grey and black) indicate the position of PCR primers used for genotyping as shown in the lower left panel; the unmodified locus gives a 1000 bp band while the CRISPR-modified locus gives a 450 bp band. WT, wild type; Het, heterozygous; Ho, homozygous. (**C**) Representative images of control and *ΔSER/ΔSER* colon tissues stained for AChE activity. Higher magnification views are shown in S9A Fig. (for FVB/N wild-type controls, n = 7 males and 6 females; for *ΔSER/ΔSER* mutants, n = 16 males and 15 females). (**D**) Representative images of βIII-Tubulin immunofluorescence labeling of e12.5 ENCCs in intestines from unsexed WT and *ΔSER/ΔSER* embryos (n = 12 FVB/N controls and 13 *ΔSER/ΔSER* mutants). Quantitative analysis of extent of ENCC colonization is shown in S9B Fig. Dashed outlines delineate the cecum (ce) and colon (co). Scale bar, 1000 μm (C); 200 μm (D).

in male *TashT*^Tg/Tg^ ENCCs (Fig 4A). Further cross-comparison of this gene set (see tabs "Unique to M-TashT (317)" and "Common to M- & F-TashT (138)" in S4 Dataset) with our prior unsexed *TashT*^Tg/Tg^-*vs*-Control RNA-seq dataset (S2 Dataset) identified 14 genes for which expression was downregulated in *TashT*^Tg/Tg^ ENCCs (Fig 4B and Table 2), as it would be expected from direct functional interaction with the *Hace1-Grik2* silencer-enriched region. From these 14 genes, *Dusp26* on chromosome 8 stood out as the best candidate (Fig 4C) based on its known phosphatase activity that inhibits p53 [48–50] and the fact that p53 can activate the expression of *DDX3X* [51, 52], the X-linked homolog of *DDX3Y*. A direct interaction between the *Dusp26* locus and the *Hace1-Grik2* silencer-enriched region was independently validated to be *TashT*-specific using chromosome conformation capture (3C)-PCR assays, which further revealed that it can occur in both male and female *TashT*^Tg/Tg^ samples at e12.5 (Fig 4D).

The discovery of a putative repressive interaction between the *Hace1-Grik2* silencer-enriched region and the p53 phosphatase coding gene *Dusp26* in *TashT*^Tg/Tg^ ENCCs prompted us to analyse p53 protein levels in these cells. As expected from the downregulation of a p53 inhibitor, immunofluorescence analyses revealed that the number of e12.5 ENCCs with high levels of p53 protein (a proxy for stabilized/activated p53 protein [53, 54]) is on average doubled in *TashT*^Tg/Tg^ embryos in comparison to *Gata4p-GFP* controls (Fig 4E). This increase seems specific to the p53 protein as *Trp53* transcript levels remain grossly unaffected by the *TashT* mutation in unsexed e12.5 ENCCs (3222 read counts in *TashT*^Tg/Tg^ *vs* 3544 read counts in *Gata4p-GFP* controls; *P* = 0.69) [29]. The increase in p53 protein levels was present in both sexes, affecting about 9% of ENCCs in *TashT*^Tg/Tg^ embryos compared to about 3–5% in *Gata4p-GFP* controls (Fig 4F). Yet, a sex-specific difference was specifically noted in *Gata4p-GFP* controls, with male embryos having half the number of ENCCs with high levels of p53 protein in comparison to female embryos (Fig 4F).

We next sought to determine whether murine *Ddx3x* and *Ddx3y*, like human *DDX3X* [51], could be directly regulated by p53. An important sub-question we also wanted to address was whether *Ddx3y* could be more sensitive than *Ddx3x* to p53 regulation, as this would help to explain why *Ddx3x* expression in ENCCs is not affected by the *TashT* mutation [29]. To begin answering these questions, we analyzed the impact of p53 knockdown on the expression of both *Ddx3x* and *Ddx3y* in three male embryonic cell lines (R1 embryonic stem cells, F9 embryocarcinoma cells, and P19 embryocarcinoma cells) via RT-qPCR. Neural crest-derived Neuro2a cells were not included in these analyses because of their lack of a Y chromosome (S10 Fig). Interestingly, expression of *Ddx3y* appeared more affected than *Ddx3x* by siRNA-mediated knockdown of p53 in both R1 and F9 cells (Fig 4G). Accordingly, chromatin immunoprecipitation (ChIP)-qPCR analyses in R1 cells further revealed a trend toward a greater transfection stress-induced interaction of p53 with the *Ddx3y* promoter in comparison to the *Ddx3x* promoter, or even the *Mdm2* promoter–a well known stress-induced p53 target [55] (Fig 4H). Collectively, these results thus support a model whereby increased p53 activation in *TashT*^Tg/Tg^ ENCCs leads to male-specific upregulation of the Y-linked gene *Ddx3y*, and not its X-linked homolog *Ddx3x*, because the *Ddx3y* promoter is more prompt to respond to p53.

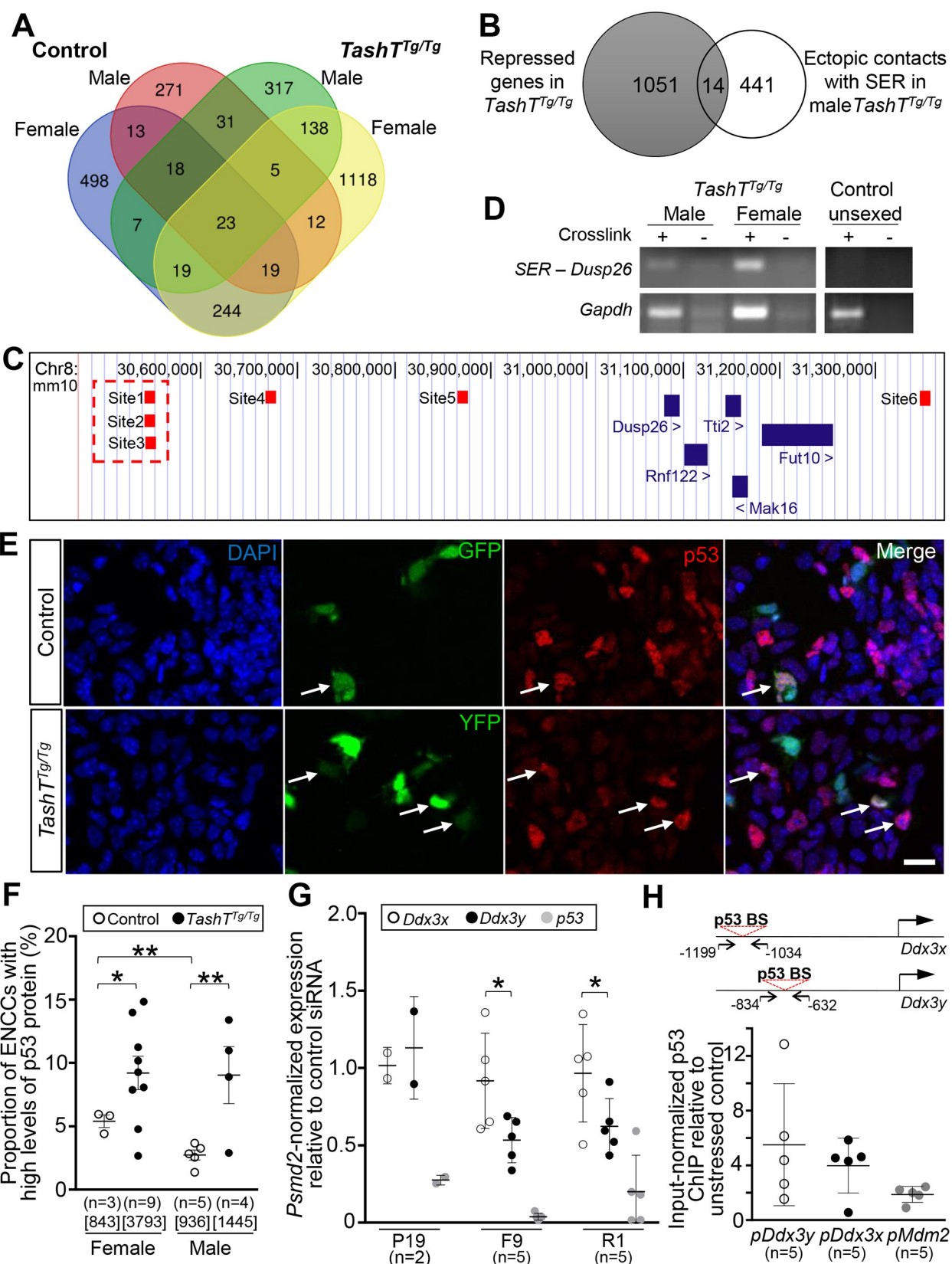

**Fig 4.** ***Ddx3y*** **upregulation may be explained by increased activity of p53 in** ***TashT*** **^{Tg/Tg} ENCCs. (A)** Venn diagram displaying all genes associated with the regions captured in the 4C-seq analysis (see Fig 3A). Shared and unique interactions are shown as a function of genotype (*Gata4p-GFP* control *vs TashT*^{Tg/Tg}) and sex (female *vs* male). The name of every involved gene is provided in S4 Dataset. **(B)** Venn diagram comparing the 1065 genes previously identified as being repressed in unsexed *TashT*^{Tg/Tg} ENCCs (S2 Dataset) with the 455 genes identified in the 4C-seq analysis as interacting with the *Hace1-Grik2* silencer-enriched region in male *TashT*^{Tg/Tg} ENCCs (317 in males only, and 138 in males and females). The 14 genes common to both sets, including the p53 phosphatase-coding gene *Dusp26*, are listed in Table 2. **(C)** Genomic position of all sequences (small red boxes, named Site1 to 6) from the *Dusp26* locus that were captured by 4C-seq (Diagram adapted from the UCSC genome browser; genome.ucsc.edu). The Site1-3 region (highlighted by a red dashed outline) was used as target sequence in the 3C-PCR experiment shown in D. **(D)** Representative 3C-PCR results showing inter-chromosomic interaction between the *Hace1-Grik2* silencer-enriched region and the *Dusp26* locus. 3C libraries (n = 3 for each condition) were made from whole e12.5 embryonic intestines dissected from sexed *TashT*^{Tg/Tg} embryos and unsexed *Gata4p-GFP* control embryos. Intra-chromosomic interaction at the *Gapdh* locus was used as library positive control, and non-crosslinked (-) cells were used as background control. **(E)** Representative confocal images of male e12.5 intestinal cells freshly dissociated from *Gata4p-GFP* control and *TashT*^{Tg/Tg} embryos, and immunolabeled for p53 (red). ENCCs are endogenously labeled with either GFP (in control) or YFP (in *TashT*^{Tg/Tg}), and those showing high levels of p53 are identified by white arrows. Scale bar, 20μm. **(F)** Sex-stratified quantitative analysis of ENCCs with high levels of p53 protein using images such as those displayed in E. The total number of counted cells is indicated in brackets. **(G)** RT-qPCR analysis of endogenous mRNA expression of *Ddx3x*, *Ddx3y* and *p53* following siRNA-mediated *p53* knockdown (relative to control siRNA) in P19, F9 and R1 embryonic cell lines. **(H)** ChIP-qPCR assays in R1 embryonic stem cells showing transfection stress-induced enrichment of p53 binding on *Ddx3y*, *Ddx3x* and *Mdm2* promoters. Relative position of the primers used to amplify *Ddx3x* and *Ddx3y* promoter sequences that contain a p53 binding site are shown at the top. (* *P*≤0.05; ** *P*≤0.01; Student's *t*-test).

## Inhibition of p53 activity eliminates the male bias in *TashT*^{Tg/Tg} embryos

To directly test the predicted p53 contribution to the male-biased severity of the *TashT* ENCC colonization defect, we turned to a pharmacological approach that previously proved to be efficient in a mouse model of Treacher-Collins syndrome–a known p53-dependent neurocristo-pathy [56]. To this end, pregnant *TashT*^{Tg/Tg} dams were administered the synthetic p53 inhibitor pifithrin-α (2 mg/kg of body weight) or vehicle only (DMSO) via daily intraperito-neal injections for 7 consecutive days, from e8.5 to e14.5. The idea was to expose vagal-derived ENCCs during their entire migration phase, from the neural tube to the distal hindgut. Embryos were then collected at e15.5 and their colon was immediately analyzed for extent of ENCC colonization without fixation, taking advantage of the endogenous YFP labeling of these cells in *TashT* embryos [29]. In accordance with our prior observations in *TashT*^{Tg/Tg} newborns [29], a statistically significant difference of about 14.5% in extent of ENCC colonization was noted between male (59.5% ± 1.8%) and female (74.0% ± 2.8) *TashT*^{Tg/Tg} embryos of the vehicle-treated group (Fig 5A and 5B). Remarkably, this difference was completely

**Table 2. List of the 14 downregulated genes in *TashT*^{Tg/Tg} ENCCs also found in the male *TashT*^{Tg/Tg} 4C-seq analysis.**

| Gene | Fold change | *TashT* mean count | Control mean count |
|:---:|:---:|:---:|:---:|
| *Lrrn2* | -2,84 | 113 | 321 |
| *Dusp26* | -2,74 | 533 | 1462 |
| *Nfasc* | -2,72 | 1696 | 4613 |
| *Prox1* | -2,56 | 142 | 365 |
| *Pnmal1* | -2,56 | 228 | 583 |
| *Hoxd3* | -2,24 | 2885 | 6454 |
| *Cntn2* | -2,22 | 161 | 357 |
| *Rbp1* | -2,21 | 6474 | 14329 |
| *Slc35e4* | -2,16 | 250 | 539 |
| *Tspan7* | -2,13 | 1359 | 2893 |
| *Bbc3* | -1,98 | 135 | 267 |
| *Mvb12b* | -1,74 | 2755 | 4811 |
| *Pik3ip1* | -1,66 | 687 | 1142 |
| *Prps2* | -1,59 | 3779 | 6011 |

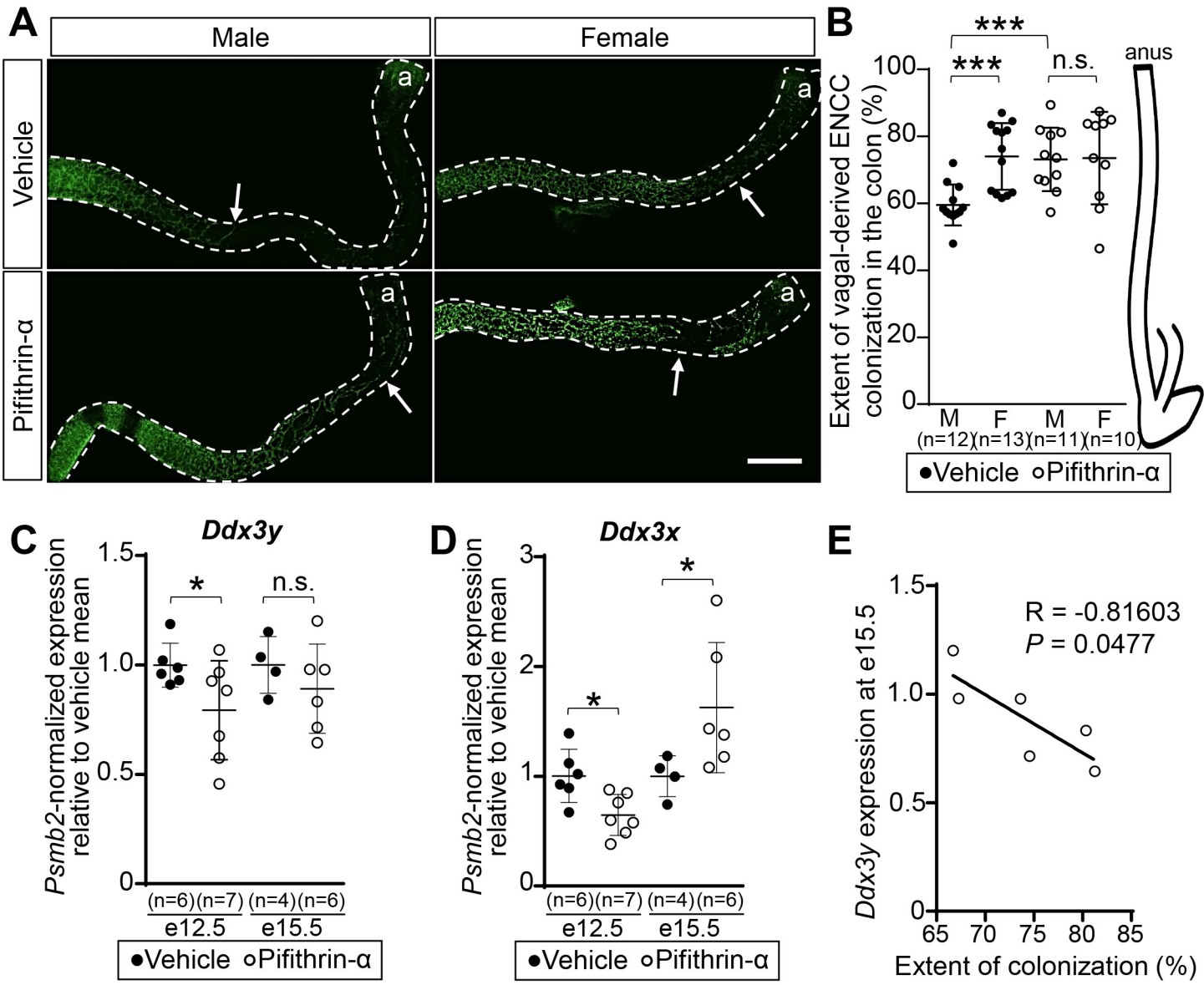

**Fig 5. *In utero* pifithrin-α treatment improves colon colonization by *TashT^{Tg/Tg}* ENCCs in a male-specific manner.** (**A**) Representative confocal images of YFP-labeled ENCCs in the colon of male and female e15.5 *TashT^{Tg/Tg}* embryos that were exposed to either pifithrin-α or only vehicle (DMSO). The white arrow indicates the migration front of vagal-derived ENCCs. Scale bar, 500 μm. (**B**) Sex-stratified quantitative analysis of vagal-derived ENCC colonization (in % of total colon length) using images such as those displayed in A. (**C-D**) RT-qPCR analysis of endogenous mRNA expression of *Ddx3y* (**C**) and *Ddx3x* (**D**) genes in *TashT^{Tg/Tg}* ENCCs recovered by FACS from male embryos exposed to pifithrin-α or only vehicle (DMSO), either during (e12.5) or after (e15.5) the treatment period. (**E**) Correlation between *Ddx3y* expression levels in ENCCs and extent of ENCC colonization in the colon for e15.5 male *TashT^{Tg/Tg}* embryos that were exposed to pifithrin-α. (R, Pearson's correlation coefficient; *$P \leq 0.05$; ***$P \leq 0.001$; n.s., not significant; Student's *t*-test).

eliminated by pifithrin-α treatment. In this group, the average extent of ENCC colonization in both male (73.1% ± 2.9%) and female (73.5% ± 4.4%) *TashT^{Tg/Tg}* embryos appeared very similar to what was observed in vehicle-treated female *TashT^{Tg/Tg}* embryos (Fig 5A and 5B).

To verify the potential involvement of *Ddx3y* in the response to pifithrin-α treatment, we evaluated its expression in male ENCCs recovered by FACS (owing to YFP fluorescence) from stomach and small intestine during the treatment period at e12.5. As observed upon siRNA-mediated knockdown of p53 expression in R1 and F9 cell lines (Fig 4G), RT-qPCR analysis

revealed a significant decrease in *Ddx3y* gene expression upon pharmacological inhibition of p53 activity (Fig 5C). When we performed the same analysis using a subset of e15.5 embryos that were previously evaluated for ENCC colonization (Fig 5A and 5B), we also detected a trend toward decreased expression of *Ddx3y* (Fig 5C) which was further found to be inversely correlated with extent of colonization (Fig 5E). In marked contrast, analysis of *Ddx3x* gene expression in response to pifithrin-α treatment gave inconsistent results in this experimental setting, showing a downregulation during the treatment period at e12.5 and an upregulation after treatment termination at e15.5 (Fig 5D). All these *in vivo* data thus strengthen the idea that DDX3Y, but not DDX3X, is likely involved in a p53-regulated molecular cascade that is responsible for the increased severity of the *TashT* ENCC colonization defect in male embryos.

## Discussion

The *TashT* mouse line is unique in being the only mouse model of HSCR that replicates the male bias observed in humans [29, 34, 37]. Here, we report that the male bias in *TashT*<sup>Tg/Tg</sup> mice involves the upregulation of the Y-linked gene *Ddx3y* in ENCCs, and that the human DDX3Y protein is also expressed in the postnatal ENS of a substantial subset of HSCR patients. In the course of our studies, we also found that overexpression of the murine ortholog of *SRY*– another Y-linked candidate gene for the HSCR male bias–has no overt impact on ENS forma- tion in mice. Our mechanistic studies further suggest that *Ddx3y* is a preferred target gene of p53, and that pharmacological inhibition of this transcription factor *in utero* is sufficient to eliminate the *TashT* male bias. This work thus identifies p53 and DDX3Y as new players in aganglionic megacolon pathogenesis in mice, a finding that might eventually be useful for the development of non-surgical therapies in humans.

Upregulation of the Y-linked *SRY* gene has been previously proposed as a pathogenic mechanism for the HSCR male bias [30]. However, this hypothesis was solely based on *in vitro* studies and protein expression data in HSCR colon tissues, without any functional assay *in vivo*. Moreover, no ENS defects were reported in mice with ubiquitous overexpression of human *SRY*, which die around two weeks of age due to multi-organ failure mainly affecting the heart, liver, lung and brain [57]. To validate the candidacy of *SRY* in the HSCR male bias, we thus decided to directly test whether mouse *Sry* can negatively impact ENS development in mice via targeted overexpression in the neural crest lineage (S6 Fig). Importantly, this work revealed no influence of *Sry* on ENS formation, even in a *TashT*<sup>Tg/Tg</sup> sensitized background (Fig 1G and 1H and S3–S5 Figs). Combined to the fact that *Sry* is not normally expressed in murine ENCCs when aganglionic megacolon develops (S1 Dataset), our negative finding thus suggests that the SRY protein is not involved in the HSCR male bias. Yet, it remains also possi- ble that SRY only affects human and not murine ENS development. Moreover, these observa- tions do not rule out a potential pathogenic role for SRY in the mature ENS, and the idea that Parkinson's disease may originate from ENS dysfunctions that propagate to the brain is espe- cially interesting in this regard [58]. Indeed, human SRY is expressed in both the postnatal ENS [30] and midbrain-associated dopaminergic neurons [59], and antisense oligonucleotide- mediated knockdown of *Sry* is neuroprotective in experimental Parkinson's disease in rats [27].

In contrast to our negative results with mouse *Sry*, several lines of evidence suggest that upregulation of the DEAD-box RNA helicase coding gene *DDX3Y* might be involved in the HSCR male bias: 1) murine *Ddx3y* is normally expressed in ENCCs when aganglionic megaco- lon develops (S2 Dataset); 2) *Ddx3y* is specifically upregulated in *TashT*<sup>Tg/Tg</sup> ENCCs (Fig 1A– 1C and S1 Dataset); 3) neural crest-directed overexpression of *Ddx3y* worsens the *TashT* male

bias in extent of aganglionosis and incidence of megacolon (Fig 1E and S1E Fig); 4) pifithrin-α-induced elimination of the *TashT* male bias in extent of aganglionosis is associated with decreased *Ddx3y* expression (Fig 5); and 5) the human DDX3Y protein is detected in enteric neurons of an important subset of HSCR patients (Fig 2 and Table 1). Of note, this latter observation challenges the view that the *DDX3Y* gene is widely transcribed, but that the protein is restricted to male germ cells [60]. However, it is important to point out that this conclusion was based on western blot analysis of only three male adult tissues (brain, kidney and testis). Since then, other groups have reported the presence of the DDX3Y protein in supplemental cell lineages, such as different types of myeloid and lymphoid hematopoietic cells [61, 62] and neurons derived from NTera2 embryocarcinoma cells [44].

The exact molecular mechanism by which DDX3Y may contribute to the development of aganglionic megacolon in a sensitized genetic background is currently unknown. The RNA helicase function of both DDX3Y and DDX3X (also known as DDX3) is believed to influence many aspects of RNA processing from transcription to translation, with an especially prominent role in the regulation of initiation of translation [63, 64]. Yet, although these proteins are highly similar (>90% sequence similarity), they are only partially redundant [65–67], meaning that DDX3Y is expected to have its own set of target transcripts. However, very little is known about the specific molecular role of DDX3Y. This situation thus makes the analysis of DDX3Y function in NTera2-differentiated neurons especially interesting for predicting DDX3Y's potential function in ENCCs. Indeed, as we observed in neural crest-derived Neuro2a cells (S1C Fig), the DDX3Y protein is enriched in the cytoplasm in NTera2-differentiated neurons, and knocking down DDX3Y in these cells impacts the translation of 71 proteins [44]. Interestingly, many of these proteins play various regulatory roles in cell migration (*e.g.*, RAB5B, RAB5C, CNN2, IFITM1, COX2, PKM1, BCAT1, SMAP1, ENO1, RAP2A, ANP32A, and STMN1), which is the main cellular process affected in *TashT*$^{Tg/Tg}$ ENCCs [29]. The fact that no transcripts other than *Ddx3y* are specifically dysregulated at the transcriptional level in male *TashT*$^{Tg/Tg}$ ENCCs (Fig 1A) supports the idea that DDX3Y mainly influences translation of migration regulatory proteins in these cells as well. As observed in NTera2-differentiated neurons [44], it can further be hypothesized that an excess of DDX3Y might result in either an increase or a decrease in translation efficiency depending of the mRNA target. Identification of these post-transcriptionally regulated mRNA targets of DDX3Y in ENCCs will require further high-throughput experiments that are beyond the scope of the current study, such as proteomics-based studies combined to RNA-binding and ribosome profiling assays. This future work should also lead to a better understanding of the very intriguing observation that DDX3Y overexpression has no impact in a female *TashT*$^{Tg/Tg}$ background. Different possibilities include the existence of either male-specific Y-linked mRNA target(s) or X-linked mRNA target(s) that escape X inactivation [68], which in both cases would make the encoded protein (s) more vulnerable to pathogenic perturbations in males. Another possibility would be that DDX3Y must act in concert with another male-specific Y-linked protein involved in initiation of translation, such as the EIF2 subunit EIF2S3Y [69, 70].

We put a lot of efforts on elucidating the molecular mechanism by which *Ddx3y* may be upregulated in male *TashT*$^{Tg/Tg}$ ENCCs, reasoning that this might eventually be useful for therapeutic purposes. This work culminated on the identification of an apparently critical role for p53, which was previously reported as a key regulator of *DDX3X* transcription in lung cancer cells [51, 52]. Our data now show that *Ddx3y* transcription can similarly be regulated by p53 in male embryonic cell lines of mouse origin, and the magnitude of this regulation even appears to be more important than for *Ddx3x* (Fig 4G and 4H). This observation is in accordance with a previous study reporting differential regulation of the *DDX3Y* and *DDX3X* promoters in various human and hamster cell lines [65], and further suggests that the importance of p53

regulation for *Ddx3x/DDX3X* expression might be context-dependent (*e.g.*, cell type- or stage-specific). In male ENCCs, p53 appears especially important for the regulation of *Ddx3y* expression, but not for the regulation of *Ddx3x* expression. *Ddx3y* expression levels always seem in phase with p53 activation levels in *TashT*$^{Tg/Tg}$ ENCCs: both are upregulated when compared to basal levels in *Gata4p-GFP* control ENCCs (*e.g.*, Fig 1C for *Ddx3y* gene expression and Fig 4E and 4F for p53 protein activity) and both are downregulated upon pifithrin-α treatment (Fig 5C). In contrast, *Ddx3x* expression levels appear either unaffected by the presence of the *TashT* mutation [29] or inconsistently modulated by pifithrin-α-mediated inhibition of p53 activity (Fig 5D). The close regulatory interaction between p53 and *Ddx3y*, combined to the detrimental impact that DDX3Y may have on colonization of male ENCCs (Fig 1E), might be one reason why p53 activity in control ENCCs appears to be lower in male than in female samples (Fig 4F). Such a safeguard mechanism would help reducing the risk that *Ddx3y* expression reaches pathogenic levels in male ENCCs.

Although we became interested in p53 because of the *TashT* mutation-mediated transcriptional downregulation of its inhibiting phosphatase DUSP26 (Fig 4B–4D and Table 2), our work does not rule out the possibility that other cellular stresses might also have contributed to the increased activation of p53 in ENCCs. Indeed, p53 can be activated by a plethora of cellular stresses [71], and increasing evidence points to inappropriate activation of p53 as the underlying cause of many developmental syndromes collectively referred to as p53-associated syndromes [72]. Most of these syndromes involve defective neural crest development [72], which appears to be due to the fact that this cell lineage is especially susceptible to p53 activation [73]. Accordingly, experimentally-induced moderate activation of p53 in either pre-migratory or migratory neural crest cells dramatically decreases their number, including those contributing to ENS formation [72]. This phenomenon appears mostly due to an increase in p53-induced apoptosis [72, 73]. However, ENCCs are known to become resistant to p53-induced apoptosis after their entry into the foregut [74], an observation that helps to explain why *TashT*$^{Tg/Tg}$ ENCCs do not have any survival deficit [29] despite showing an increase in p53 activation (Fig 4E and 4F). Altogether these observations thus support a general model whereby many different HSCR-causing mutations might trigger p53 activation in ENCCs and lead to a preferential increase of *DDX3Y* expression in males (along with its detrimental effects on ENCC colonization), without affecting canonical apoptosis-associated p53 target genes. Although it remains challenging to test this gene defect-independent hypothesis with the relevant human cell type, this can now be envisaged thanks to recent methods for *in vitro* differentiation of ENS progenitors from pluripotent stem cells (using either ES or iPS cells) [75, 76].

If confirmed, the gene defect-independent and p53-dependent model mentioned above would explain why human genetic studies have constantly failed to identify gene variants specifically associated with the HSCR male bias [14]. Importantly, our finding that the negative impact of the putative p53-DDX3Y axis on ENCCs can be reversed by pharmacological means in mice also extends the list of potential mutation-independent preventive interventions for HSCR. Downregulating p53 activity during ENS development *in utero* might thus be considered an option for male fetuses in families with high genetic risk, just like the suggestion of avoiding nutritional deficiencies and high doses of specific medicines [77]. Considering that the vast majority of HSCR patients display aganglionosis over only a short segment of the distal colon [1, 2], such intervention might well be sufficient to prevent megacolon occurrence in many male individuals (up to 50% of them according to our human DDX3Y immunofluorescence data; Fig 2 and Table 1). While it is reasonable to believe that p53 inhibition increases tumorigenesis risk, this risk is minimized by the fact that such intervention would only need to be done over a relatively short period of time (*i.e.*, for a maximum of 3–4 weeks, between weeks 4 to 7 of gestation). Alternatively, future work might help to identify a cellular stressor

upstream of p53 that could also be pharmacologically targeted, as reported in a mouse model of Treacher-Collins syndrome [78].

## Materials and methods

### Ethics statement

Work with mice was approved by the Comité institutionnel de protection des animaux (CIPA reference #650 and #975) of Université du Québec à Montréal (UQAM), and performed in accordance with the guidelines of the Canadian Council on Animal Care (CCAC). For euthanasia, isoflurane-anesthetized mice were exposed to gradual-fill carbon dioxide.

Families of human participants provided written informed consent on studies, which were approved by the institutional human research ethics committees of the Université du Québec à Montréal (Comité institutionnel d'éthique de la recherche avec des êtres humains; protocol # 491) and the Centre hospitalier universitaire Sainte-Justine (Comité d'éthique à la recherche; protocol # 4172).

### Animals

Heterozygous *Gata4p-GFP* and homozygous *TashT* transgenic lines were maintained and genotyped as previously described [29, 40]. $_{Myc}Ddx3y$- or $_{Myc}Sry$-overexpressing mice were generated via standard pronuclear microinjection [79] into FVB/N zygotes. In each case, a *Tyrosinase* minigene [31] was co-injected to allow visual identification of transgenic animals via rescue of pigmentation [80], and homozygous animals were confirmed by backcrossing with wild-type FVB/N mice. For both $_{Myc}Ddx3y$ and $_{Myc}Sry$ transgenes, relevant coding sequences were subcloned in frame with an N-terminal Myc tag into an in house-modified pIRES2-EGFP vector (Clontech), where the CMV promoter was replaced by a synthetic promoter composed of the *Hsp68* minimal promoter [43] and neural crest-specific enhancers from *Sox10* (*U3*) [41] and *Ednrb* [42]. *Ddx3y* coding sequences and *Ednrb* regulatory sequences were amplified by RT-PCR and PCR, respectively, using nucleic acids from an e12.5 male FVB/N embryo, and then cloned in pGEM-T vector and fully sequenced to confirm the absence of PCR-caused mutations. The *domesticus Sry* cDNA (ORF and partial 3' UTR), *Sox10 U3* enhancer, and *Hsp68* minimal promoter were subcloned from vectors kindly provided by Dr Chris Lau (University of California San Diego, USA), Dr Michael Wegner (Friedrich-Alexander-Universität Erlangen-Nürnberg, Germany) and Dr Rashmi Kothary (Ottawa Hospital Research Institute, Canada), respectively.

For CRISPR/Cas9-mediated deletion at the *Hace1-Grik2* locus, double-stranded DNA oligonucleotides coding for top-scoring sgRNAs (as determined using the *crispr.mit.edu tool)* that flank the silencer-enriched region were cloned into the pX330 vector (Addgene plasmid # 42230; gift from Dr Feng Zhang) [81], and resulting constructs were used for *in vitro* transcription as previously described [82]. A mixture of *Cas9* mRNA (100 ng/μl) and both sgRNAs (50 ng/μl for each) was then prepared and used to microinject FVB/N zygotes, using conventional techniques [79]. Resulting *Hace1-Grik2$^{ΔSER}$* mice were identified by PCR using primers located outside or within the deletion, and the exact nature of the deletion was determined by sequencing the amplicons generated with flanking primers. Details about all cloning and genotyping oligonucleotide primers can be found in S1 Table.

Embryos were obtained by timed pregnancies, with noon of the day of vaginal plug detection being considered e0.5. For *in utero* inhibition of p53, cyclic pifithrin-α hydrobromide (Sigma Cat.#P4236) was administered to pregnant *TashT$^{Tg/Tg}$* dams via intraperitoneal injections of a 0.4 mg/ml solution (in 2% DMSO-PBS) at a final concentration of 2 mg per kg of mouse body weight, as previously described [56]. Control animals were administered 2%

DMSO-PBS only. Dams were treated once a day (around noon) for a period of 7 days between e8.5 and e14.5, and embryos were collected at e15.5 for analysis.

## Sex-specific transcriptome profiling

Ribosomal RNA-depleted RNA-seq librairies were both generated (from 300ng of total RNA) and sequenced by McGill University and Génome Québec Innovation Centre using the HiSeq 2000 platform (Illumina). For both male and female *TashT*$^{Tg/Tg}$ ENCCs, three biological replicates were generated from pools of e12.5 embryos, which were sexed based on morphological differences between male and female gonads. Cell dissociation of *TashT*$^{Tg/Tg}$ bowel (stomach and small intestine), FACS-mediated recovery of ENCCs, and extraction of total RNA were all performed as previously described [29]. Paired-end sequences (100-bp in length; between 58–150 million reads per replicate) were mapped onto the mm10 *Mus musculus* reference genome, and analyzed for differential gene expression using DESeq and edgeR packages.

## RT-qPCR

For quantitative RT-PCR analysis of ENCCs or transfected cell lines, cells of interest were recovered by FACS and their total RNA content was extracted as previously described [83]. Total RNA (100 ng per sample) was reverse transcribed using the Superscript III enzyme and a mixture of poly-dT18 and random primers, in accordance with the manufacturer's instructions (ThermoFisher Scientific). 2 μl of the resulting cDNA pool was then used for amplification of the desired target using specific primers. Expression levels of the housekeeping gene *Psmd2* were used for normalization. The qPCR program consisted of 45 cycles of 10 s at 95˚C and 30 s at 60˚C, with a melting curve from 65˚C to 95˚C. Relative expression was calculated with the 2-ΔΔCt method that refers to the ratio of the target genes to the housekeeping gene [84]. Details about qPCR primers can be found in S1 Table.

## Human studies

Human specimens of ENS-containing sigmoid colon (corresponding to the "transition" zone) were obtained from two different HSCR cohorts without genetic information. The first cohort is described in Soret *et al*. [32] as the HSCR case group (n = 12; 10 boys and 2 girls; aged between 17 and 454 days at the time of surgery). The second cohort was constituted from four patients (4 boys; aged between 22 and 349 days at the time of surgery) who underwent surgical resection at the Centre hospitalier universitaire Sainte-Justine (Montreal, Canada). Details for each analyzed tissue are displayed in Table 1. Tissues were processed for immunofluorescence as previously described [32], using mouse monoclonal anti-DDX3Y (Sigma Cat. #WH0008653M1; dilution factor 1/500) and guinea pig polyclonal anti-PGP9.5 (Neuromics Cat.#GP14104; dilution factor 1/100) antibodies.

## 4C-seq and 3C-PCR

4C-seq was performed essentially as previously described [46], with slight modifications to accommodate limited starting material (1x10$^6$ FACS-recovered ENCCs per sample) and a different sequencing platform for longer reads (Ion PGM; ThermoFisher Scientific). For each sample, captured chromatin fragments were reduced in size by sonication using an E220 Focused-ultrasonicator (Covaris) set with a peak power of 175, a duty factor of 10%, and 200 cycles for 45 seconds. Sequencing libraries were prepared using the Ion Xpress Plus Fragment Library Kit and the gDNA Fragment Library Preparation protocol (ThermoFisher Scientific Cat.#MAN0009847). Sequencing particles were generated with the Ion PGM Hi-Q Chef Kit

(ThermoFisher Scientific Cat.#MAN0010919), and sequencing was performed on Ion 318 Barcoded chips adapted for sequencing fragments of 400 bp or less (ThermoFisher Scientific Cat. #4488146). The 4C-seq analysis was performed using two biological replicates per group. All data from both replicates were kept for further analysis, recognizing that chromatin contacts are often transitory [85] and most especially during development [86, 87]. Sequencing data were then analyzed using the w4Cseq pipeline [47], which was modified to accommodate long single reads instead of short paired-end reads. Genes associated with captured fragments were identified using the Table Browser tool from the UCSC Genome Browser (genome.ucsc.edu). 3C-PCR was performed as previously described [29], using primers listed in S1 Table.

## Staining of developing and postnatal murine ENS

Staining of AChE activity in postnatal colon and whole-mount immunofluorescence staining of e12.5 bowel tissues were both performed using standard approaches, as previously described [32, 88]. Antibodies used for whole-mount immunofluorescence included mouse monoclonal anti-βIII-Tubulin clone 2G10 (Abcam Cat.#ab78078; dilution factor 1:500), rabbit polyclonal anti-GFP (Abcam Cat.#Ab290; dilution factor 1:500) and rat polyclonal anti-SOX10 (generated by MediMabs for the Pilon lab; dilution factor 1:500). Analysis of endogenous YFP fluorescence in unfixed freshly dissected e15.5 $TashT^{Tg/Tg}$ colons was performed as previously described [29]. For p53 and Myc tag immunofluorescence in dissociated tissues, acute cultures of e12.5 bowel were prepared and stained essentially as previously described [83]. Briefly, for each tested embryo, dissected stomach and small intestine were dissociated in 100 μL of EMEM containing collagenase (0.4 mg/ml; Sigma Cat.#C2674), dispase II (1.3 mg/ml; ThermoFisher Scientific Cat.#17105–041) and DNAse I (0.5 mg/ml; Sigma Cat.#DN25) at 37˚C. Cells were then centrifuged at 300g for 3 minutes, resuspended in 200 μl of EMEM with 10% FBS on a μ-slide 8-well Ibitreat plate (Ibidi 80826) coated with fibronectin (250 μg/ml; EMD Millipore Cat.#FC010), and incubated at 37˚C with 5% CO2 for 4 to 6 hours. Cells were finally fixed for 10 minutes in 4% PFA on ice, and processed for immunofluorescence with mouse monoclonal anti-p53 (Cell Signaling Cat.#2524; dilution factor 1:250), rabbit polyclonal anti-GFP (Abcam Cat #Ab290; dilution factor 1:500), mouse monoclonal anti-Myc tag (In house hybridoma clone 9E10; dilution factor 1:100) and/or rat polyclonal anti-SOX10 (generated by MediMabs for the Pilon lab; dilution factor 1:500).

## Cell culture and transfections

Neuro2a neuroblastoma cells, F9 embryocarcinoma cells, P19 embryocarcinoma cells, and R1 embryonic stem cells were maintained and propagated as previously described [89, 90]. To evaluate the activity of $_{Myc}Ddx3y$-IRES-eGFP and $_{Myc}Sry$-IRES-eGFP transgenic constructs, 3x10⁵ Neuro2a cells were seeded in 6-well plates containing coverslips and transfected with Genejuice reagents (Novagen) in accordance with manufacturer's instructions. Coverslips were then used to analyze transfected cells for co-expression of eGFP and Myc-tagged proteins via anti-Myc immunofluorescence (mouse monoclonal Myc tag antibody clone 9E10; in house hybridoma; dilution factor 1:100). To evaluate the importance of p53 in the regulation of *Ddx3x* and *Ddx3y* gene expression, male cell lines (F9, P19 and R1) were seeded in 100mm plates (1x10⁶ to 2x10⁶ cells) and transfected with either *p53* (stealth siRNA *Trp53*; Cat. #1320001, MSS212108; ThermoFisher Scientific) or negative control (stealth RNAi siRNA negative control; Cat.#12935300 Med GC; ThermoFisher Scientific) siRNAs using jetPRIME reagents (Cat.#114–75; Polyplus transfection) in accordance with the manufacturer's instructions. Each siRNA (30nM) was co-transfected with empty pIRES2-EGFP plasmid (10μg; Clontech), and transfected cells (EGFP+) were then recovered by FACS 48h later and processed for

RT-qPCR analysis of *p53*, *Ddx3x* and *Ddx3y* gene expression. Primer details can be found in S1 Table.

## Image acquisition and processing

Images were acquired either with a Leica M205 FA stereomicroscope equipped with a Leica DFC 495 camera (for staining of AChE activity in postnatal colon tissues and fluorescence imaging in whole embryos) or a Nikon A1 laser scanning confocal microscope (for fluorescence imaging in general). Image processing and analysis was all done with the ImageJ software.

## ChIP analysis

ChIP assays in R1 embryonic stem cells were performed as previously described [83], using a mouse monoclonal p53 antibody (Cell Signaling #2524; dilution factor 1:200). To activate p53, R1 cells were subjected to a transfection stress prior to ChIP, by transfecting the empty pIRES2-EGFP vector as described above for siRNA analyses. Occupation of p53 binding sites from *Ddx3y*, *Ddx3x* and *Mdm2* promoters in control and stressed R1 cells was determined by qPCR, using the program described in the RT-qPCR section. ChIP efficiency was calculated by comparison of input-normalized values (in %) obtained with stressed *vs* control cells. PCR-amplified regions containing p53 binding sites were determined based on the literature [51] and a MatInspector analysis (www.genomatix.de) for the *Ddx3x* and *Ddx3y* promoters, or the literature only [55] for the *Mdm2* promoter.

## Statistics

Where applicable, data are presented as mean ± standard deviation, with the number of biological replicates indicated in figures and/or legends. Student's *t*-test (two-tailed) was used to determine statistical significance, using 0.05 as cut-off value for *P*. Numerical data and statistics for all graphs can be found in S2 Table.

## Supporting information

**S1 Fig. Characterization of the $_{Myc}$Ddx3y transgenic mouse line.** (**A**) Complete sequence of the *Ednrb-Sox10-Hsp68* synthetic promoter. Restriction sites used for cloning are in lowercase letters. (**B**) Schematic representation of the $_{Myc}$Ddx3y-IRES-eGFP transgenic construct, with regulatory sequences in grey. (**C**) Immunofluorescence labeling of Neuro2a cells transfected with the $_{Myc}$Ddx3y-IRES-eGFP transgenic construct and showing co-expression of $_{Myc}$Ddx3y (red) and GFP (green) proteins. Scale bar, 10μm. (**D**) RT-qPCR analysis of $_{Myc}$Ddx3y transgene expression in e12.5 *TashT*$^{Tg/+}$;$_{Myc}$Ddx3y$^{Tg/+}$ ENCCs recovered by FACS. (**E**) Detailed quantitative analysis of the length of ENS-covered colon (in % of total colon length) in P18-P50 *TashT*$^{Tg/Tg}$ mice bearing or not a transgenic allele from each of the three $_{Myc}$Ddx3y transgenic lines. Every $_{Myc}$Ddx3y transgenic line is identified using the same color code as in Fig 1E and 1F. The dashed lines show the previously described threshold level interval above which megacolon is less likely to occur in FVB/N mice [29]. (* $P \leq 0.05$; ** $P \leq 0.01$; Student's *t*-test). (TIF)

**S2 Fig. Detection of $_{Myc}$DDX3Y protein in e12.5 midgut cells dissociated from *TashT*$^{Tg/+}$; Ddx3y$^{Tg/+}$ embryos.** Representative confocal images of e12.5 midgut cells dissociated from *TashT*$^{Tg/+}$ control and *TashT*$^{Tg/+}$;$_{Myc}$Ddx3y$^{Tg/+}$ (line 3) embryos, and immunolabeled for SOX10 (red) and Myc tag (grey). ENCCs are endogenously labeled with YFP owing to the

*pSRYp[1.6kb]-YFP* transgene in the *TashT* line. Scale bar, 20 μm.
(TIF)

**S3 Fig. Overexpression of either $_{Myc}$Ddx3y or $_{Myc}$Sry has no impact on ENS patterning.** Representative images of proximal colon tissues collected from P18-50 *TashT*$^{Tg/+}$ mice bearing or not $_{Myc}$*Ddx3y* or $_{Myc}$*Sry* transgenic constructs and stained for AChE activity. Scale bar, 1000 μm.
(TIF)

**S4 Fig. Overexpression of $_{Myc}$Ddx3y, but not $_{Myc}$SRY, worsens extent of aganglionosis in TashT$^{Tg/Tg}$ males.** Representative images of P18-50 colon tissues collected from male mice and stained for AChE activity. Corresponding quantitative analyses are shown in Fig 1E and 1G. Scale bar, 1000 μm.
(TIF)

**S5 Fig. Overexpression of either $_{Myc}$Ddx3y or $_{Myc}$SRY has no impact on extent of aganglionosis in TashT$^{Tg/Tg}$ females.** Representative images of P18-50 colon tissues collected from female mice and stained for AChE activity. Corresponding quantitative analyses are shown in Fig 1F and 1H. Scale bar, 1000 μm.
(TIF)

**S6 Fig. Characterization of the $_{Myc}$Sry transgenic mouse line. (A)** Schematic representation of the $_{Myc}$*Sry-IRES-eGFP* transgenic construct, with regulatory sequences in grey. **(B)** Immunofluorescence labeling of Neuro2a cells transfected with the $_{Myc}$*Sry-IRES-eGFP* transgenic construct and showing co-expression of $_{Myc}$SRY (red) and GFP (green) proteins. Scale bar, 10μm. **(C)** Representative image of GFP-labeled neural crest-derived cells in e11.5 $_{Myc}$*Sry-IRES-eGFP* transgenic embryos (n = 12). Left panels show GFP labeling of cranial nerves and dorsal root ganglia. Right panel is a high magnification view of the gut, to show GFP labeling of ENCCs in the prospective stomach (St) and small intestine (SI). Scale bar, 1000 μm (left panel); 200 μm (Right panel).
(TIF)

**S7 Fig. Activity of the *Ednrb-Sox10U3-Hsp68* synthetic promoter in ENCCs. (A-B)** Representative confocal images of e12.5 midguts collected from $_{Myc}$*Sry*$^{Tg/+}$ embryos and immunolabelled for SOX10 (red), βIII-Tubulin (grey), GFP (green) and DAPI (blue). High magnification views shown in B are delineated by dashed square in A. As evidenced on both x-y and x-z planes, GFP signal is mainly present in SOX10$^+$ cells. **(C)** Quantification of the number of SOX10$^+$ GFP$^+$ double positive cells on the total number of SOX10$^+$ cells. The total number of counted cells is indicated in brackets. Scale bar, 50 μm (A) and 20μM (B).
(TIF)

**S8 Fig. Selection of the best primer pair for the 4C-seq analysis. (A)** Genomic representation of the *Hace1-Grik2* intergenic region on chromosome 10, using the UCSC genome browser (genome.ucsc.edu) and its vertebrate Multiz Alignment & Conservation tract. Conserved elements (CE) are indicated by either black (neutral activity) or red (silencer activity) boxes, based on transcriptional activity in luciferase assays previously performed in Neuro2a and P19 cell lines [29]. Position of 4C-seq primer pairs is illustrated in blue (1), yellow (2), purple (3), green (4), and pink (5). **(B)** A test 4C-library was prepared using Neuro2a cells and used to evaluate efficiency of each primer pairs. The third pair (purple) was selected based on PCR profile and proximity to silencer elements.
(TIF)

**S9 Fig. The colon of *Hace1-Grik2^ΔSER/ΔSER* embryos is normally colonized by ENCCs.** (**A**) Representative images of P18-50 proximal colon tissues collected from wild-type (WT; FVB/N control) and *Hace1-Grik2^ΔSER / ΔSER* (*ΔSER / ΔSER*) mice, and stained for AChE activity. Lower magnification images are displayed in Fig 3C. (**B**) Quantitative analysis of the extent of colon colonization by βIII-Tubulin⁺ ENCCs (in % of total colon length) in WT and *Hace1--Grik2^ΔSER/ΔSER* e12.5 embryos, showing normal colonization in mutant animals. Representative images are shown is Fig 3D. Scale bar, 1000 μm.
(TIF)

**S10 Fig. Sexing of Neuro2a cell line.** (**A**) PCR-based sexing of Neuro2a cells showing the presence of the X-linked gene *SmcX* (300 pb) but the absence of Y-linked genes *Zfy* (420 pb) and *SmcY* (280 pb). Genomic DNA from female (F) or male (M) mice was used as controls. (**B**) Relative expression of *Psmd2* (autosomal gene), *Ddx3x* (X-linked gene) and *Ddx3y* (Y-linked gene) in Neuro2a compared to a known male mouse cell line (P19). Normalization to *Psmd2* expression reveal the absence of *Ddx3y* expression.
(TIF)

**S1 Table. List of all primers used in this study.** Full sequence of all oligonucleotides used in this study is shown in the 5' to 3' orientation. The purpose for which oligonucleotides were used is also mentioned.
(PDF)

**S2 Table. Numerical data and statistics for all graphs.** Each graph has its own tab.
(XLSX)

**S1 Dataset. Differential gene expression in male *vs* female *TashT^Tg/Tg* ENCCs (>1.5-fold change; *P*≤0.05; minimum of 250 reads in at least one condition).**
(XLSX)

**S2 Dataset. Differential gene expression in *TashT^Tg/Tg* vs *Gata4p-GFP* control ENCCs (>1.5-fold change; *P*≤0.05; minimum of 250 reads in at least one condition).**
(XLSX)

**S3 Dataset. List of all loci and associated genes captured by 4C-seq.** Each sample has its own tab.
(XLSX)

**S4 Dataset. Lists of genes for all Venn diagram categories displayed in Fig 4A.** Each category has its own tab.
(XLSX)

## Acknowledgments

The authors thank Normand Lapierre and Guillaume Bernas (Transgenesis core, CERMO-FC, UQAM) for the production of genetically-modified mice, Denis Flipo (Cellular analyses core, CERMO-FC, UQAM) for assistance with confocal imaging and cell sorting, Geneviève Bourret (Genomics core, CERMO-FC, UQAM) for assistance with preparation of 4C-seq libraries, Golrokh Kiani (Bioinformatics core, CERMO-FC, UQAM) for assistance with analysis of 4C-seq data, as well as Dr Ann Aspirot and Dr Natalie Patey for access to human samples (CHU Ste-Justine). The Massively parallel sequencing platform as well as the Bioinformatics platform of the McGill University and Génome Québec Innovation Center are also thanked for transcriptome sequencing and analyses of sequencing results, respectively. Likewise, the Genomics platform of the Centre de recherche du Centre Hospitalier Universitaire Sainte-Justine as well

as the Bioinformatics platform of the Institut de Recherches en immunologie et en cancérologie (IRIC) are thanked for sequencing and analyses of 4C-seq libraries, respectively. The Genomics core facility at the Sunnybrook research institute is also thanked for the second round of sequencing of 4C-seq libraries.

## Author Contributions

**Conceptualization:** Nicolas Pilon.

**Formal analysis:** Tatiana Cardinal, Karl-Frédérik Bergeron, Nicolas Pilon.

**Funding acquisition:** Nicolas Pilon.

**Investigation:** Tatiana Cardinal, Karl-Frédérik Bergeron, Ouliana Souchkova, Amélina Guillon.

**Methodology:** Tatiana Cardinal.

**Resources:** Rodolphe Soret, Christophe Faure.

**Supervision:** Nicolas Pilon.

**Writing – original draft:** Tatiana Cardinal, Nicolas Pilon.

**Writing – review & editing:** Karl-Frédérik Bergeron, Nicolas Pilon.

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
