## [Decision Letter · Decision Letter 0]

23 Oct 2019

Dear Dr Pilon,

Thank you very much for submitting your Research Article entitled 'Male-biased aganglionic megacolon in the TashT mouse model of Hirschsprung disease is caused by upregulation of the p53-DDX3Y regulatory axis' to PLOS Genetics. Your manuscript was fully evaluated at the editorial level and by independent peer reviewers. The reviewers appreciated the attention to an important problem, but raised some substantial concerns about the current manuscript. In particular there were concerns by the reviewers and editorial members in several areas:

1 - regarding the expression analyses that led to selection of Ddx3y there needs to be normalization of mutants to sex specific controls and greater explanation provided regarding the observed over expression of mesenchymal markers.

2 - regarding the EdnrB/Sox10 enhancer construct, greater detail about this tool (including final sequence and expression relative to known ENS markers) needs to be provided to instill confidence in it's validity.

3 - regarding the MycDdx3y-IRES-eGFP transgene additional effort should be made to show that Ddx3y protein is made from this construct, that the construct does not alter the patterning or composition of the ENS in transgenic mice, and the detail of which transgenic line (1, 2, or 3) were used to generate the data in subsequent figures within the manuscript should be included in the body of the text. 

Based on the comments, we will not be able to accept this version of the manuscript, but we would be willing to review again a much-revised version. We cannot, of course, promise publication at that time.

If you decide to revise the manuscript for further consideration at PLOS Genetics, please aim to resubmit within the next 60 days, unless it will take extra time to address the concerns of the reviewers, in which case we would appreciate an expected resubmission date by email to plosgenetics@plos.org.

[LINK]

We are sorry that we cannot be more positive about your manuscript at this stage. Please do not hesitate to contact us if you have any concerns or questions.

Yours sincerely,

Michelle Southard-Smith

Guest Editor

PLOS Genetics

Gregory Barsh

Editor-in-Chief

PLOS Genetics

Reviewer's Responses to Questions

**Comments to the Authors:**

Reviewer #1: In this manuscript, Cardinal et al. aim to determine molecular mechanisms underlying the male-biased “Hirschsprung disease”-like phenotype of the TashT mouse model. Hirschsprung disease (HSCR) is characterized by the lack of enteric neurons in a variable range of the colon, and has a male bias. Mouse models of currently known major risk genes do not replicate this particular feature. The authors have previously provided a HSCR model, TashT, where an insertional enhancer mutation results in Enteric Neural Crest transcriptome dysregulation. In the present manuscript, Ddx3y is identified as the only gene upregulated in both male-female; TashT-control comparisons. Authors provide a novel construct allowing efficient expression in ENCC; Ddx3y overexpression in TashT transgenic mice accentuates the phenotype. Ddx3y may be relevant in the human context as it is specifically expressed in enteric neurons of male HSCR patients, however, whether this expression differ from the general population of males is not addressed. Through a series of elegant experiments (high-throughput 4C-seq, elimination of TashT-locus, and siRNA-mediated knockdown and more) a new model for the male-sex-bias is presented. Improper TashT chromatin contacts lead to downregulation of a p53 phosphatase; the resulting increased p53 activity enhances Ddx3y expression.

The identification of Ddx3y as the causative male bias of HSCR in the TashT mouse model, may present a new target for future pharmacological prevention of HSCR in families with high genetic risk. The manuscript is overall well-written and organized. All conclusions are sound and appropriately discussed.

Major Issues:

The study presents a combined Ednr/Sox10 enhancer construct to drive expression in the developing ENS. As this construct is new and previously uncharacterized, please provide data showing its efficiency and specificity in the developing gut (as well comment on the expression elsewhere in the embryo). An attempt is made for the mycSRY-IRES-eGFP construct, but a correlation to ENCCs at a cellular level is needed. Please provide percentage of Sox10 cells expressing the transgene, and the level of specificity within the gut. A validated ENCC enhancer-construct for expression in mouse ENS will be appreciated in the field at large.

Minor issues:

Introduction (row 71-73): The description of HSCR as loss of ganglia from distal colon is simplified. Approximately 20% of HSCR patients display long-segment or total colonic aganglionosis. A small adjustment of this sentence is required to account for these cases.

Reviewer #2: PGENETICS-D-19-01494

In this manuscript, the authors report identification of P53 and DDX3Y as genes upregulated and potentially related to the male bias of HSCR. The goal of identification of genes/gene products that underlie the male bias in the genesis of HSCR is an important step in developing treatment strategies not dependent upon surgery. Identification of the Ddx3y, a Y-linked gene, as the causal gene for male bias in the TashT mouse line is potentially very significant. Of more interest, is the finding that p53-DDX3Y regulatory outcomes are reversible and possibly identifies the (a) pathogenic mechanism for male bias HSCR in the TashT line. The transcriptomic approach to identification of differentially expressed genes in male vs female is excellent. It should be noted that this manuscript is very well written and the data beautifully presented. Comments and/or concerns are listed below

1. The finding that DDX3Y protein was only detected in 50% of male HSCR patients is somewhat problematic. It is curious that methods, other than immunofluorescence were not used to try to detect this protein. What is the percentage of expression of DDX3Y in different lines of mice? This result needs a bit more attention in the discussion.

2. Although the studies designed to assess potential differences between Ddx3x and Ddx3y in their regulation by P53 are important, the use of the cell lines chosen makes interpretation of these data difficult. Gene regulation in each of these lines is not “normal.” Each of these lines has multiple issues related to gene expression especially of neural crest related genes. It would be interesting to repeat this experiment in sorted ENCCs which can be efficiently grown in culture and which are amenable to knock-down. Alternatively, ENS-derived ENCCs efficiently develop embryoid bodies in culture.

3. Inasmuch as all male mice did not develop HSCR in response to altered expression of Ddx3y, it is an overstatement that this gene is responsible for the male bias of HSCR in all mice; please tone this down.

Reviewer #3: This manuscript uses elegant and state-of-the-art techniques to elucidate the genetic basis for male-biased aganglionosis and megacolon in the TashT mouse model of HSCR. Building on their prior work, the authors identify a novel locus Ddx3y that is upregulated in male TashT mice and that this directly interferes with the p53 protein pathway.

However, the enthusiasm of this reviewer was dampened by some of the major issues identified in the analyses, experimental design and the resulting inference presented by the authors in this manuscript. These are as follows:

Major Issues:

1. The authors wrote the methods section that frequently required the reviewer to refer back to their earlier reports. Example: “Analyses of endogenous YFP fluorescence was performed as previously described.”. This required the reviewer to refer back to their earlier data and understand how the techniques that were central and crucial to understanding the experiments in this manuscript were performed.

2. On referring back to the earlier reports, the reviewer found that the YFP fluorescence in the TashT line in colon but not in small intestine is also expressed in the mesenchymal cells. Indeed the data presented in this manuscript shows that in the TashT vs Control gene expression differences from the YFP/GFP+ ENCC from small intestine and stomach (Dataset S2), the TashT transgenic animals have an overexpression of mesenchymal markers (Acta2, Actg2, Myh11) compared to controls. They also include genes such as Lumican and Col6a4 that are important for the normal migration of ENCC. Now this either means that the fidelity of the sort is compromised and that the TashT dataset contains contaminating genes from YFP- non-ENCC or that TashT causes some kind of fate-transition of these neural crest-derived cells towards a mesenchymal fate. In any case, this dataset is the basis of identification of the gene expression differences between normal and transgenic animals and these important issues have not been addressed adequately. The reviewer is careful not to critique the past reports and is confining himself/herself to this report where this data has been presented and is central to the identification of Ddx3y in the next screen.

3. The intriguing part of the datasets presented here and that have not been commented on is that the TashT animals have an overall increase in expression of Gdnf but an underexpression of Ret and Gfra1 as compared to controls.

4. In addition, the sex-biased TashT gene expression analyses has not been normalized each sex of mutant mice to their own controls. An ideal and correct experiment would be to perform TashT males vs Control males and TashT females vs Control females and then compare the significantly different genes between these two datasets. In the current experimental design and analyses, there would be an expected domination of X and Y linked genes and the study becomes underpowered and important genes that are present on autosomes and are differentially regulated between males and females do not get tested. The authors are advised to perform these analyses so that the transgenic associated sex-differences can be baselined correctly.

5. As a control for Ddx3y (Fig 1C), the authors chose Eif2s3y, a gene that is significantly upstream and is in the opposite direction. The authors do not provide any rationale for why this gene was chosen instead of the gene Uty that is immediately next to Ddx3y and is in the same direction.

6. The authors used three transgenic lines of Ddx3y overexpression in the TashT background and a cumulative analyses shows a significant male biased aganglionosis and megacolon (Fig 1 E, F). However, Fig S1 show that there are major variations between the three lines. While all three lines show significant differences compared to WT (Fig S1C), only the second line shows any significant difference in aganglionosis compared to the TashT background. The reviewer then wonders why the other lines were included in the main data analyses or why the Lines 1 and 3 that show a significant overexpression of Ddx3y fail to show significant increase in the length of aganglionosis (Fig S1D). This combined with the data that shows that overexpression of Ddx3y in females had no effect makes the reviewer wonder whether Ddx3y is really that central to the male bias.

7. The authors implicate sex-specific trans-activation of p53 is in the male-biased phenotype. However, the controls themselves show what seems to be a significant reduction in p53 expression (Fig 4F). This is very important but this is neither statistically tested by the authors nor the implications of this tested and discussed.

8. Fig 5 shows that the addition of a p53 inhibitor to the Ddx3y – TashT mice abrogates the male bias. However, the correct way of analyzing this would be to compare each drug-treatment to its own sex-specific control. Analyzing it in that manner shows that there is a large variation and that the drug treatment is either subclassifying the effect into two populations (where one population has an effect and other doesn’t) or that the drug dosage and timing isn’t optimized.

9. The authors should also present what the Ddx3y immunoreactivity in the aganglionic part of the murine and human gut look like.

Minor issues:

1. DataS4: the nomenclature of Unique to and Common to is confusing and doesn’t describe whether genes in those datasets are specific to one or other groups.

2. Line 185: The authors should say “associated with” rather than “responsible”

**Have all data underlying the figures and results presented in the manuscript been provided?**

Reviewer #1: Yes

Reviewer #2: Yes

Reviewer #3: Yes

PLOS authors have the option to publish the peer review history of their article (what does this mean?). If published, this will include your full peer review and any attached files.

Reviewer #1: Yes: Ulrika Marklund

Reviewer #2: No

Reviewer #3: No

---

## [Decision Letter · Decision Letter 1]

17 Jun 2020

Dear Dr Pilon,

Thank you very much for revising your prior submission of the Research Article titled 'Male-biased aganglionic megacolon in the TashT mouse model of Hirschsprung disease is caused by upregulation of the p53-DDX3Y regulatory axis' to PLOS Genetics. Your revised manuscript was fully evaluated at the editorial level and by three independent peer reviewers. The reviewers appreciated the effort you put into the revision the additional details that clarified multiple aspects, unfortunately some concerns remain to be addressed.

1 - As currently presented, the manuscript implies that DDX3Y is the only sex-associated transcript that exhibits differential expression. If the manuscript is to retain this stance, comparing the TashT males vs Control males and TashT females vs Control females is an important aspect that is should be done and incorporated. Please take this aspect into consideration when revising the manuscript. If current circumstances prevent this analysis, one alternative is to restructure your presentation in the text so that this statement is toned down.

2- As previously commented upon in the initial reviews the manuscript infers that there is a causal link between the change in DDX3Y expression on p53 inhibition and the reduction/absence in aganglionosis in the TashT mouse. Definitive support for this statement could be generated by showing that there is no change to aganglionosis when p53 is inhibited in the TashT-DDX3Y over expression mouse. Alternatively, you could revise the text and the paper title to remove this over-interpretation of current information and indicate that your results suggest an interaction that could be investigated in the future.

If you decide to revise the manuscript for further consideration at PLOS Genetics, please aim to resubmit within the next 60 days, unless it will take extra time to address the concerns of the reviewers, in which case we would appreciate an expected resubmission date by email to plosgenetics@plos.org.

[LINK]

We are sorry that we cannot be more positive about your manuscript at this stage. Please do not hesitate to contact us if you have any concerns or questions.

Yours sincerely,

Michelle Southard-Smith

Guest Editor

PLOS Genetics

Gregory Barsh

Editor-in-Chief

PLOS Genetics

Reviewer's Responses to Questions

**Comments to the Authors:**

Reviewer #1: The revised version include the sequence of the EdnrB/Sox10 construct and quantification of the proportion of Sox10 cells where the construct is active. For transparency I only ask for two more details:

1) Please add the individual data-points in your bar-graph in Figure S7. Although not mentioned in the first review, it is highly advisable to include data-points in all graphs included in the manuscript.

2) Please add a picture of a transverse cut through the intestine showing the expression of YFP/SOX10/BIIITub to indicate the specificity of expression within the ENS.

Reviewer #2: This study provides strong evidence that the p53-DDX3Y regulatory axis is at least partially responsible for the male bias in Hirschsprung’s disease (HSCR). This is a very interesting and potentially important finding; the authors have identified a potential treatment target for subsets of patients with HSCR. This data are clearly presented and the studies are well controlled. The authors have done an excellent job of addressing all of the issues raised in the previous review. This manuscript will make a significant contribution to the ENS field in general and to those most interested in HSCR.

Reviewer #3: The authors in this revised manuscript have addressed many of the comments put forth by the reviewer in the prior review to varying degrees of satisfaction. The reviewer will address the remaining issues in order.

1. The reviewer thanks the authors for significantly re-writing the manuscript to put forth their point of how the sex-stratified analyses of TashT mice was done and how the identification of the gene DDX3Y could have been made even with the strategy suggested by this reviewer. While the reviewer does not disagree that DDX3Y could have been identified in the other manner as well, the reviewer would humbly suggest that the suggested strategy would provide more and clearer understanding of the genetic factors altered in the TashT mice that make male mice more susceptible to aganglionosis. The reviewer would refer the authors to their Fig 4f, which now shows that the expression of p53 gene is significantly different in an analyses of (female control to female TashT) and (male control to male TashT). As a result, with the analyses the reviewer suggested, the authors would have easily picked up p53 as a gene that is expressed more than 2 fold differently between the TashT mice of the two sexes. While the current analyses is fine for identification of DDX3Y, it gives an erroneous impression that only this gene is significantly different and hence is the only key gene implicated in male bias in TashT aganglionosis. Such an analyses also might hold the key to explaining why DDX3Y overexpression only causes aganglionosis in the TashT background, which is the question posed by the authors in their Discussion.

2. The reviewer congratulates the authors on showing strong data on the effect of p53 inhibition on aganglionosis in the TashT mouse line. However, the reviewer feels that the causal axis of p53 and DDX3Y proposed by the authors is tenuous, since the authors did not check what would happen with p53 inhibition to aganglionosis when DDX3Y is overexpressed. The experiment to really nail the inference that "Male-biased aganglionic megacolon in the TashT mouse model of Hirschsprung disease involves upregulation of the p53-DDX3Y regulatory axis" (which is the title of the manuscript) would require the authors to show that there is no change to aganglionosis when p53 is inhibited in the TashT-DDX3Y overexpression mouse. This would provide clear and unequivocal evidence that there is a p53-DDX3Y regulatory axis that is key to the male bias of HSCR associated aganglionosis.

3. As a minor point, the reviewer would advise the authors to streamline their manuscript to allow for a better and a more focused read.

**Have all data underlying the figures and results presented in the manuscript been provided?**

Reviewer #1: Yes

Reviewer #2: Yes

Reviewer #3: Yes

PLOS authors have the option to publish the peer review history of their article (what does this mean?). If published, this will include your full peer review and any attached files.

Reviewer #1: No

Reviewer #2: No

Reviewer #3: No

---

## [Decision Letter · Decision Letter 2]

23 Jul 2020

Dear Dr Pilon,

We are pleased to inform you that your manuscript entitled "Male-biased aganglionic megacolon in the TashT mouse model of Hirschsprung disease involves upregulation of p53 protein activity and Ddx3y gene expression" has been editorially accepted for publication in PLOS Genetics. Congratulations!

Yours sincerely,

Michelle Southard-Smith

Guest Editor

PLOS Genetics

Gregory Barsh

Editor-in-Chief

PLOS Genetics

Comments from the reviewers (if applicable):

Reviewer's Responses to Questions

**Comments to the Authors:**

Reviewer #1: I am satisfied with the last editions. Congratulations to a nice manuscript.

Reviewer #3: The authors have responded satisfactorily to the reviewers comments.

**Have all data underlying the figures and results presented in the manuscript been provided?**

Reviewer #1: Yes

Reviewer #3: Yes

PLOS authors have the option to publish the peer review history of their article (what does this mean?). If published, this will include your full peer review and any attached files.

Reviewer #1: No

Reviewer #3: No

**Data Deposition**

http://datadryad.org/submit?journalID=pgenetics&manu=PGENETICS-D-19-01494R2

**Press Queries**

---

## [Editor Report · Acceptance letter]

3 Sep 2020

PGENETICS-D-19-01494R2 

Male-biased aganglionic megacolon in the TashT mouse model of Hirschsprung disease involves upregulation of p53 protein activity and Ddx3y gene expression 

Dear Dr Pilon, 

We are pleased to inform you that your manuscript entitled "Male-biased aganglionic megacolon in the TashT mouse model of Hirschsprung disease involves upregulation of p53 protein activity and Ddx3y gene expression" has been formally accepted for publication in PLOS Genetics! Your manuscript is now with our production department and you will be notified of the publication date in due course.

With kind regards,

Jason Norris

PLOS Genetics

On behalf of:
